# HiMoLE: Towards OOD-Robust LoRA via Hierarchical Mixture of Experts

**Yinuo Jiang**[1,2],**Yan Xiaodong**[5], **Keyan Ding**[3], **Deng Zhao**[5],

**Lei Liang**[5],**Qiang Zhang**[2,4*], **Huajun Chen**[1,2*]

[1]College of Computer Science and Technology, Zhejiang University
[2]ZJU-Ant Group Joint Lab of Knowledge Graph
[3]ZJU-Hangzhou Global Scientific and Technological Innovation Center, Zhejiang University
[4]ZJU-UIUC Institute, Zhejiang University
[5]Ant Group
{qiang.zhang.cs, huajunsir}@zju.edu.cn

## Abstract

Parameter-efficient fine-tuning (PEFT) methods, such as LoRA, have enabled the efficient adaptation of large language models (LLMs) by updating only a small subset of parameters. However, their robustness under out-of-distribution (OOD) conditions remains insufficiently studied. In this paper, we identify the limitations of conventional LoRA in handling distributional shifts and propose **HiMoLE** (**Hi**erarchical **M**ixture of **Lo**RA **E**xperts), a new framework designed to improve OOD generalization. HiMoLE integrates hierarchical expert modules and hierarchical routing strategies into the LoRA architecture and introduces a two-phase training procedure enhanced by a diversity-driven loss. This design mitigates negative transfer and promotes effective knowledge adaptation across diverse data distributions. We evaluate HiMoLE on three representative tasks in natural language processing. Experimental results evidence that HiMoLE consistently outperforms existing LoRA-based approaches, significantly reducing performance degradation on OOD data while improving in-distribution performance. Our work bridges the gap between parameter efficiency and distributional robustness, advancing the practical deployment of LLMs in real-world applications.

## 1 Introduction

Large language models (LLMs) have brought transformative advances across a wide range of domains. However, their unprecedented scale incurs substantial computational and storage costs. To address this issue, parameter-efficient fine-tuning (PEFT) techniques, such as LoRA [1], have emerged as practical solutions. By updating only a small subset of model parameters, PEFT methods reduce storage and computational requirements while achieving performance comparable to full-model fine-tuning. This efficiency makes them especially attractive for real-world deployments.

Despite these advantages, PEFT methods face a critical shortcoming: **limited generalization under out-of-distribution (OOD) conditions**. While deep learning models typically perform well on in-distribution (ID) data, their performance often degrades when faced with data that deviates from the training distribution [2, 3]. This issue persists even in LLMs after full fine-tuning and is particularly pronounced in domains characterized by high heterogeneity [4, 5], such as biomedicine and the

---

[*]Corresponding authors.

39th Conference on Neural Information Processing Systems (NeurIPS 2025).

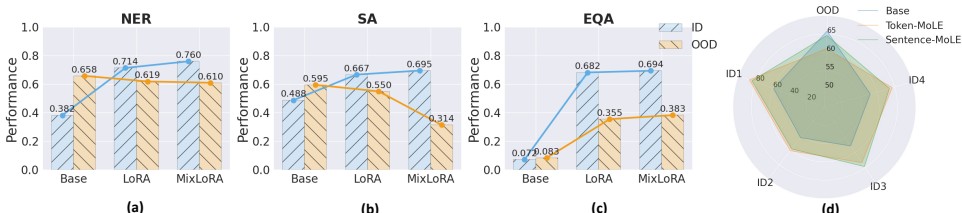

Figure 1: Robustness analysis of parameter-efficient fine-tuning. (a)–(c): In-distribution (ID) and out-of-distribution (OOD) results for the *Base*, *LoRA*, and *MixLoRA* models on three representative tasks: Named Entity Recognition (NER) in biomedicine, Sentiment Analysis (SA) in social science, and Extractive Question Answering (EQA) in general domain. (d): Impact of routing granularity in Mixture-of-LoRA-Experts, where "OOD" refers to an OOD validation dataset, while "ID1" "ID2" "ID3" and "ID4" represent the four ID validation datasets. Token-level routing yields better ID performance but fails to generalize to OOD data. In contrast, sentence-level routing improves OOD robustness at the cost of ID accuracy.

social sciences. Surprisingly, although LoRA and related PEFT methods have gained widespread adoption, their robustness to distributional shifts remains largely unexplored. Our empirical analysis (Fig. 1(a)-(c)) reveals that standard LoRA suffers considerable accuracy drops when applied to tasks requiring adaptation across diverse knowledge domains. These findings suggest intrinsic limitations in LoRA's ability to generalize beyond the training distribution, motivating the need for more robust PEFT strategies.

Mixture-of-parameter-efficient-expert (MoPE) methods attempt to improve generalization by integrating the Mixture-of-Experts (MoE) framework with PEFT, demonstrating effectiveness in multi-task settings [6, 7, 8, 9]. However, their effectiveness under OOD conditions remains underexplored. As shown in Fig. 1(a)(b), both LoRA and MixLoRA suffer notable performance degradation in knowledge-intensive tasks (e.g., Named Entity Recognition (NER) in biomedicine and Sentiment Analysis (SA) in social science) when evaluated on OOD data. As shown in Fig. 1(c), in general-domain tasks (e.g., Extractive Question Answering (EQA)), the gap between ID and OOD performance persists, revealing limited robustness. In some cases, MixLoRA even underperforms standard LoRA, suggesting potential overfitting. We identify a key source of this limitation: token-level routing in MoPE models is prone to expert misallocation under distributional shift. Local token-level features often fail to capture the global semantics necessary for robust generalization, resulting in brittle routing decisions in unseen contexts. These observations motivate the central question of our study: **How can parameter-efficient fine-tuning methods be improved to enhance in-distribution performance while also ensuring robustness to out-of-distribution data?**

To address this issue, we propose **HiMoLE** (**Hi**erarchical **M**ixture **o**f **L**oRA **E**xperts), a novel framework that introduces structural sparsity and hierarchical design into the LoRA architecture. HiMoLE extends conventional MoPE models via the hierarchical architecture which manifests in two dimensions: hierarchical expert design and hierarchical routing strategy. To further improve knowledge utilization and reduce redundancy, we introduce a two-phase training scheme augmented with a diversity-promoting loss. In summary, our work makes the following key contributions:

- **Empirical diagnosis of OOD limitations in LoRA.** We systematically investigate the generalization performance of LoRA under distributional shift, revealing significant weaknesses in its ability to transfer across heterogeneous domains.

- **A novel hierarchical MoPE framework.** We propose HiMoLE, which introduces hierarchical expert architectures and routing strategies into the PEFT paradigm. This structure mitigates negative transfer and promotes positive transfer, offering a new direction for improving PEFT robustness under distributional shift.

- **Theoretical and empirical validation.** We provide theoretical insights into the advantages of hierarchical routing under distributional shift. Experiments across multiple domains show HiMoLE improves OOD generalization while maintaining strong ID performance.

## 2 Background and Related Work

### 2.1 OOD Generalization

Out-of-distribution generalization is essential for deploying language models in real-world scenarios, where data distributions are inherently diverse, non-stationary, and unpredictable [10, 11]. This need is especially pronounced in high-stakes domains such as clinical decision support and social science analytics, where knowledge continuously evolves and data often deviate from training distributions. In such contexts, models must demonstrate robustness to emergent semantic patterns, novel entity relationships, and shifting contextual dependencies. Despite the remarkable progress of large language models across a wide range of tasks and benchmarks, recent studies [4, 12, 13] have revealed significant vulnerabilities under distributional shifts. These findings expose the limitations of current fine-tuning strategies and underscore the urgent need for methods explicitly designed to enhance OOD robustness.

### 2.2 Mixture of Parameter-efficient Experts

**Parameter-Efficient Fine-Tuning (PEFT)**    As the scale of LLMs continues to grow, PEFT has emerged as a practical and cost-effective adaptation strategy. PEFT techniques update only a small subset of model parameters—such as adapter layers or low-rank matrices—while keeping the majority of the pre-trained model frozen [14, 15, 16]. A widely adopted PEFT method is LoRA [1], which inserts trainable low-rank adapters into pre-trained layers and updates them during fine-tuning. LoRA achieves competitive performance with a substantially reduced memory footprint. However, PEFT often struggles to generalize across new distributions. The limited number of trainable parameters can restrict the model's capacity to adapt to distributional shifts or novel task requirements.

**Integrating MoE with PEFT (MoPE)**    To reconcile scalability and efficiency, recent work proposes integrating MoE with PEFT techniques, resulting in the MoPE paradigm. In MoPE, each expert is instantiated using a PEFT configuration (e.g., LoRA), and a routing module dynamically assigns inputs to appropriate experts. This hybrid design seeks to combine the modular adaptability of MoE with the resource efficiency of PEFT. MoPE methods differ in routing granularity. Token-level routing operates at the sub-sentence level. For example, MixLoRA [6] combines multiple LoRA experts with a shared FFN and incorporates an auxiliary load balancing loss to mitigate expert usage imbalance. LoRAMoE [17] employs a router network to reduce knowledge forgetting. HydraLoRA [9] adopts an asymmetric architecture with a shared LoRA $A$ matrix and expert-specific $B$ matrices. In contrast, sentence-level routing mechanisms operate at the input sentence level. MOELoRA [18] performs explicit task-to-expert assignment using task metadata, deterministically routing input sentences based on task identifiers. MOCLE [7] clusters instruction semantics and activates task-specific experts by assigning inputs to their corresponding instruction cluster. Although MoPE frameworks have demonstrated effectiveness in multi-task and in-distribution settings, their robustness under distributional shifts remains largely unexplored. This motivates the need for architectures specifically designed to handle complex domain shifts and enhance OOD generalization.

## 3 Method

### 3.1 Preliminaries

**Formulation of OOD Generalization.**    Let $x$ denote the input data and $y$ the output. Out-of-distribution generalization refers to scenarios where the test distribution $P_{\text{test}}(x, y)$ differs from the training distribution $P_{\text{train}}(x, y)$, while preserving core semantic relationships. This work focuses on the joint occurrence of two distributional shifts [19]:

1. Covariate Shift: The input distribution changes ($P_{\text{test}}(x) \neq P_{\text{train}}(x)$), but the conditional distribution remains invariant($P_{\text{test}}(y \mid x) = P_{\text{train}}(y \mid x)$).
2. Concept Shift: The input-conditional distribution changes( $P_{\text{test}}(y \mid x) \neq P_{\text{train}}(y \mid x)$ ), which may arise from label semantics or task definitions evolving across domains.

Robust generalization in this setting requires models to: (1) handle divergent input distributions (covariate shift), and (2) adapt to latent conceptual variations (concept shift).

**Identification of OOD Generalization Problem in PEFT**    Following the protocol established by [4], we consider three key criteria for identifying OOD data: (1) diverse data sources, (2) low SimCSE similarity [20] with the in-distribution dataset, and (3) measurable performance degradation in models. However, in practice, we find that the second criterion—low SimCSE similarity—is not always reliable, especially in knowledge-dense domains. Fine-tuned LLMs can still generalize effectively even when SimCSE scores are low. As such, we primarily rely on criteria (1) and (3) in our OOD dataset selection. Here, performance degradation refers to reduced model performance on OOD data relative to ID data, which can manifest in two ways: (1) the model performs worse than its pre-fine-tuned counterpart, and (2) the performance gain on OOD data is substantially smaller than that on ID data. We fine-tune models using LoRA on each task's ID dataset and evaluate them on both ID and OOD test sets. As shown in Fig. 1(a)(b)(c), LoRA fails to significantly enhance OOD robustness. To further illustrate this, we analyze failure cases in a biomedical NER task. The fine-tuned model demonstrates two main types of errors in OOD data: (1) failure to produce outputs in the correct structured format, and (2) mislabeling of general-domain entities—for instance, tagging symptoms as diseases (see Appendix A for detailed case studies).

**Probing OOD Robustness in MoPE**    Prior research [21, 22] on neural network optimization has shown that sparsely activated architectures often generalize better than dense ones, owing to dynamic parameter specialization and reduced task interference. In LLMs, the MoE framework embodies this principle by leveraging conditional computation through expert routing, enabling scalable and efficient learning. Building on these insights, we investigate the OOD robustness of various MoPE configurations, focusing on token-level and sentence-level routing strategies. As illustrated in Fig. 1(d), our results highlight a key trade-off between routing granularities: **Token-level routing** attains state-of-the-art performance on ID data by exploiting fine-grained contextual cues but exhibits pronounced OOD performance degradation, suggesting overfitting to surface-level patterns. **Sentence-level routing** offers improved OOD robustness by aligning with global semantics but suffers from reduced ID performance due to its limited sensitivity to local details. This finding highlights the necessity of developing a unified approach that can effectively integrate both routing granularities, achieving high ID accuracy while maintaining strong generalization across OOD scenarios.

### 3.2   Architecture of HiMoLE

In this subsection, we introduce HiMoLE, a Hierarchical Mixture of LoRA Experts model designed to flexibly address the OOD robustness challenges inherent in LoRA-based fine-tuning. An overview of the HiMoLE architecture is shown in Fig. 2(a).

**Hierarchical Experts**    Our parameter-efficient expert architecture is organized into $N$ Knowledge Competition Groups (KCGs), each consisting of $M$ Knowledge Collaboration Experts (KCEs). Since domains with heterogeneous knowledge comprise multiple distinct subdomains, we assign each KCG to a unique subdomain and initialize its parameters by fine-tuning on the corresponding sub-dataset (as detailed in Section 3.3, Training Strategy). All KCEs within a KCG share the same initialization, providing a consistent foundation of subdomain-specific knowledge. Each KCE is implemented as a LoRA module, formally defined as:

$$E = BA, A \in \mathbb{R}^{d_{\text{in}} \times r}, B \in \mathbb{R}^{r \times d_{\text{out}}}, r \ll \min(d_{\text{in}}, d_{\text{out}}). \tag{1}$$

The experts interact through three distinct modes. (1) Intra-group collaboration: KCEs within the same KCG specialize in subdomain-specific patterns while leveraging shared knowledge, enabling efficient adaptation and positive transfer. (2) Cross-group competition: Different KCGs compete to route tokens to the most relevant KCEs, thereby reducing interference across subdomains and mitigating negative transfer. (3) Cross-group collaboration: KCEs across different KCGs may cooperate to improve generalization, promoting knowledge reuse and transferability. This structured interplay between competition and collaboration ensures both specialization and synergistic learning across knowledge boundaries.

**Hierarchical Routing Strategy**    The core of HiMoLE lies in its hierarchical routing strategy, which integrates sentence-level and token-level expert selection to achieve domain-aware and adaptive inference. For a given input sentence, we first compute a sentence-level representation denoted by $h_{\text{sen}}$ via applying average pooling over the token-level hidden states $h_{\text{token}}$. This pooled representation

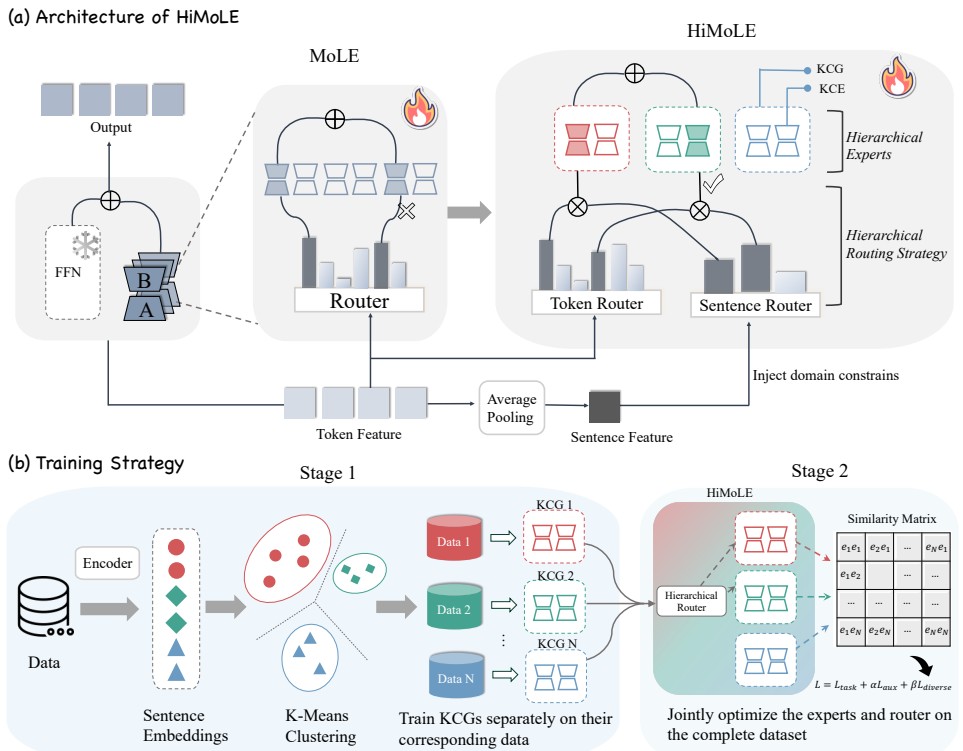

Figure 2: Illustration of the proposed HiMoLE. (a) Architecture of HiMoLE. Unlike MoLE, HiMoLE combines hierarchical experts—organized as Knowledge Competition Groups (KCGs) and their internal Knowledge Collaboration Experts (KCEs)—with a hierarchical routing strategy that performs sentence-level coarse allocation followed by token-level refinement. This architecture is designed to enhance OOD robustness in LoRA-based fine-tuning. (b) Training strategy of HiMoLE, which adopts a two-stage training strategy: first, each KCG is trained independently on a clustered sub-dataset; second, both the expert parameters and hierarchical routing components are jointly optimized.

is processed by a sentence-level router $f_{\text{sen}}(\cdot)$ (implemented as a linear layer parameterized by $W_{\text{sen}}$), to compute the initial allocation scores $G_{\text{sen}}$ over KCGs:

$$G_{\text{sen}} = f_{\text{sen}}(h_{\text{sen}}) = W_{\text{sen}} \cdot h_{\text{sen}}. \tag{2}$$

These sentence-level scores serve as a coarse-grained guide, determining which KCGs are most relevant to the input. Subsequently, for each token in the sentence, a token-level router $f_{\text{token}}(\cdot)$ (implemented as a linear layer parameterized by $W_{\text{token}}$) refines this allocation score by integrating the token-specific hidden state $h_{\text{token}}$:

$$G_{\text{token}} = f_{\text{token}}(h_{\text{token}}) = W_{\text{token}} \cdot h_{\text{token}}. \tag{3}$$

The final gating weights matrix $G_{\text{hie}}$ are computed using a softmax-normalized fusion of $G_{\text{sen}}$ and $G_{\text{token}}$, followed by top-$k$ selection to ensure sparsity:

$$G_{\text{hie}} = \text{KeepTop-}k\left(\text{Softmax}\left(G_{\text{sen}} \odot G_{\text{token}}\right)\right). \tag{4}$$

The forward process of the HiMoLE layer replaced the traditional FFN layer can be represented as:

$$o = W_0 \cdot h_{\text{token}} + \sum_{i=1}^{N \times M} G_{\text{hie}}^{(i)} \cdot E_i \cdot h_{\text{token}}, \tag{5}$$

where $W_0$ is the parameter matrix of the original FFN layer of the LLM, and $o$ denotes the output. The scalar $G_{\text{hie}}^{(i)}$ modulates the contribution weight of the $i$-th expert $E_i$. We provide the definitions of the symbols in Appendix B.

In summary, the hierarchical routing mechanism enables HiMoLE to balance domain specialization and generalization. Sentence-level routing assigns inputs to suitable KCGs based on coarse semantic

cues, while token-level routing fine-tunes expert selection for dynamic, context-aware feature fusion. This design allows for flexible expert collaboration and competition, ultimately enhancing OOD robustness and domain-adaptive inference.

## 3.3 Training Strategy

As shown in Fig. 2(b), we adopt a two-stage training strategy to construct and optimize the HiMoLE framework. The first stage initializes the Knowledge Competition Groups (KCGs) with specialized domain knowledge, while the second stage jointly optimizes both the expert networks and the hierarchical routing mechanisms.

**Stage 1: Initializing Knowledge Competition Groups**  We begin with the assumption that each task may span multiple subdomains. To capture this diversity, we partition the training dataset into $N$ subsets, each corresponding to a distinct semantic cluster, and train $N$ KCGs in parallel. To perform data clustering, we first use a pre-trained encoder to obtain semantic embeddings for each data instance. We then apply the $K$-means clustering algorithm to group the data into $N$ clusters. Each cluster is treated as a sub-dataset, and a separate KCG is independently trained on it. This process results in $N$ distinct groups of LoRA-based LLM experts, each specialized in a specific knowledge subdomain.

**Stage 2: Co-optimizing the Experts and Routers**  After initializing the $N$ KCGs, we jointly optimize the expert parameters and the hierarchical routing modules. To encourage diversity among experts and reduce redundancy, we introduce a diversity loss $\mathcal{L}_{\text{diverse}}$. Let $e_n$ denote the output of the $n$-th KCG, computed as:

$$e_n = \sum_{m=1}^{M} G_{\text{token}}^{(m)} \cdot E_m \cdot h_{\text{token}}, \quad \text{where} \quad E_m \in \text{KCG}_n. \tag{6}$$

We normalize each expert output as: $e_n \leftarrow \frac{e_n}{\max(\|e_n\|_2, \epsilon)}$, where $\epsilon$ is a very small number such as $10^{-8}$. We then compute pairwise cosine similarities $S_{nl} = \langle e_n, e_l \rangle$ among all the KCG pairs, and define the diversity loss as the average similarity across all unique expert pairs:

$$\mathcal{L}_{\text{diverse}} = \frac{1}{N(N-1)} \sum_{n=1}^{N} \sum_{l=1, l \neq n}^{N} S_{nl}. \tag{7}$$

To control computational cost, $\mathcal{L}_{\text{diverse}}$ is computed every ten layers using a sampled subset of expert outputs. The final training objective combines task loss $\mathcal{L}_{\text{task}}$, auxiliary loss $\mathcal{L}_{\text{aux}}$, which is employed to mitigate the unbalanced load for experts (following [23], see Appendix C for details), and the diversity loss:

$$\mathcal{L} = \mathcal{L}_{\text{task}} + \alpha \mathcal{L}_{\text{aux}} + \beta \mathcal{L}_{\text{diverse}}. \tag{8}$$

## 3.4 Theoretical Analysis of Generalizability in Sparse Routing Systems

We analyze how hierarchical expert routing reduces generalization error by mitigating gradient conflicts through structured sparsity in expert selection. Let $t$ index tokens and $\theta$ be the parameters of an expert $E$, which includes the low-rank matrices $A$ and $B$ as defined in Eq. 1. Let $\nabla_\theta \mathcal{L}(\cdot)$ denote the gradient of the loss with respect to $\theta$. We define the expected pairwise gradient similarity as follows:

$$SimGrad := \mathbb{E}_{t \neq t'} \left[ \cos \left( g_t, g_{t'} \right) \right], \quad \text{where} \quad g_t := \nabla_\theta \mathcal{L}(h_t). \tag{9}$$

**Definition 1 (Gradient Conflict)** *Let $h_t \neq h_{t'}$ be inputs from distinct tokens. A gradient conflict occurs if* $\cos \left( \nabla_\theta \mathcal{L}(h_t), \nabla_\theta \mathcal{L}(h_{t'}) \right) < 0$.

In this context, a higher value of $SimGrad$ indicates better alignment between token gradients and thus fewer gradient conflicts [24].

**Theorem 1 (Hierarchical Routing Mitigates Gradient Conflicts)** *Let* $SimGrad_{hie}$ *and* $SimGrad_{token}$ *denote the expected pairwise gradient similarity under hierarchical and token-only*

*routing, respectively. Then hierarchical routing with composition $f_{sen}(h_{sen}) \odot f_{token}(h_{token})$ yields:*

$$SimGrad_{hie} = SimGrad_{token} + \Delta_\theta, \tag{10}$$

*where $\Delta_\theta >= 0$. Proof. See Appendix D.*

That is, hierarchical routing reduces the prevalence of conflicting gradients by inducing structural sparsity in expert selection. We then connect gradient alignment to generalization through a bound on the generalization error.

**Lemma 1 (Generalization Bound via Gradient Variance [25])** *Let $g_t := \nabla_\theta \mathcal{L}(h_t)$, and define the gradient variance as $V(g) := \mathbb{E}\left[||(g_t - \mathbb{E}(g_t)||^2\right]$. Then, the generalization error of stochastic gradient descent with additive Gaussian noise satisfies:*

$$\text{Gen} \leq \sqrt{\frac{R^2}{b} \sum_{\tau=1}^{T} \frac{\eta_\tau^2}{\sigma_\tau^2} \mathbb{E}\left[V(g)\right]}, \tag{11}$$

*where $R$ represents a constant related to the properties of the loss function and the data distribution, and $b$ denotes the number of training samples. $T$ is the total number of iterations. $\eta_\tau$ is the learning rate at step $\tau$, and $\sigma_\tau$ is the standard deviation of the Gaussian noise at step $\tau$.*

**Theorem 2 (Gradient Conflict Reduction Enhances Model Generalization)** *Assume hierarchical routing achieves a lower gradient variance such that $V(g_{hie}) <= V(g_{token})$, then under the conditions of Lemma 1, hierarchical routing yields a tighter generalization bound, i.e., $\text{Gen}_{hie} \leq \text{Gen}_{token}$.*

*Proof.* By monotonicity of the square root and the inequality on gradient variance, the result follows directly from Lemma 1. See Appendix D for further discussion.

## 4 Experiments

### 4.1 OOD benchmark

**Name Entity Recognition (NER).**  To emulate real-world data heterogeneity and enhance the complexity of the NER task, we selected the biomedical domain as our experimental scenario—a knowledge-intensive field characterized by diverse entity types and lexical variations. For the construction of the ID dataset, we rigorously curated English-language resources from the BigBio benchmark [26], with the corpora primarily sourced from PubMed Central (PMC), a premier repository of peer-reviewed biomedical literature. This process resulted in BigBio-NER, the largest dataset for biomedical NER. For the selection of OOD datasets, we adopted the criteria outlined in Section 3.1, choosing the rare disease dataset [27] sourced from the National Organization for Rare Disorders database [28].

**Sentiment Analysis (SA).**  To further enhance the complexity of the SA task and better simulate real-world application scenarios, we frame our SA experiments within social science contexts, where affective expressions exhibit heightened subjectivity and domain-specific connotations. We adopted the sentiment analysis component in SOCIALITEINSTRUCTIONS [29] dataset as our ID dataset. This comprehensive collection of socially-oriented textual interactions contains sentiment labels across various social science scenarios. For the selection of OOD data, we adopted the criteria outlined in Section 3.1 and chose the OPTIMISM [30] dataset.

**Extractive Question Answering (EQA).**  Following previous work [4], we chose SQuAD [31] as the ID dataset, which constructs question-answer pairs based on Wikipedia passages. For the selection of OOD data, we chose NewsQA [32], which writes questions for CNN news articles, each of which requires reasoning to answer.

### 4.2 Experimental Settings

**Base Model and Data Separation**  For the NER task, we employ OneKE-13B [33] as our base model, which is capable of generalized knowledge extraction across multiple domains and tasks. For

Table 1: Comparative performance of different LoRA methods under out-of-distribution scenarios. Please refer to Appendix E.3 for the metrics details and Appendix E.5 for the complete results. The best results on ID data and the best results on OOD data excluding the base model are highlighted in **boldface** and underlined, respectively.

| Task | NER | | | | | | SA | | | EQA | | | |
|------|-----|---|---|---|---|---|----|---|---|-----|---|---|---|
| Dataset | ID | | | OOD | | | ID | OOD | | ID | | OOD | |
| Metric | F1 | P | R | F1 | P | R | EM | REM | EM | EM | ROUGE-2 | EM | ROUGE-2 |
| Base Model | 52.5 | 56.2 | 51.7 | 65.8 | 63.1 | 76.4 | 48.8 | 59.5 | 56.9 | 7.2 | 12.3 | 8.3 | 15.8 |
| LoRA | 73.9 | 77.5 | 73.3 | 61.9 | 56.0 | 77.8 | 66.7 | 55.7 | 55.0 | 68.2 | 48.7 | 35.5 | 28.2 |
| MixLoRA | 76.0 | 78.2 | 75.7 | 61.0 | 59.1 | 73.7 | 69.5 | 66.5 | 31.4 | 69.4 | 48.6 | 38.3 | 27.9 |
| HydraLoRA | 77.3 | 78.4 | 76.7 | 62.9 | 59.4 | 75.5 | 70.3 | 67.2 | 30.4 | 68.7 | 46.4 | 38.3 | 26.4 |
| **HiMoLE** | **77.9** | **78.7** | **77.3** | 65.3 | 64.4 | 74.3 | **73.3** | 68.8 | 32.8 | **70.5** | **49.8** | 38.9 | 28.7 |

the SA and EQA tasks, we utilize Llama-2-7B [34] as our base model. To separate in-distribution data, we use BioBERT [35], BERTweet [36], and DeBERTa-v3-large [37] as encoders to extract sentence-level embeddings for the NER, SA and EQA tasks, respectively. Through sentence feature-based $K$-means clustering, we obtain subsets of sizes 4, 3 and 3 for each task (see the visualization of the clustering results in Appendix E.1).

**Baselines and Settings**   We compare our HiMoLE method against LoRA and traditional mixture of LoRA experts. In our HiMoLE approach, each Knowledge Component Group consists of 4 Knowledge Component Experts. For the mixture of LoRA experts, we compared state-of-the-art methods, including MixLoRA [6] and HydraLoRA [9]. In all mixture of experts methods, we adopted the tok-$k$ routing and set $k$ in tok-$k$ to 2, while setting the LoRA rank $r$ to 8. We apply MOE to the FFN layer of every Transformer block. To maintain uniformity in parameter sizes across all methods, we set the rank $r$ to 80 in traditional LoRA.

### 4.3   Primary Results

In Table 1, we provide a comprehensive comparison of HiMoLE with various baselines. There are several observations: (1) Overall, HiMoLE achieves the best performance across all tasks on both ID and OOD datasets compared to other LoRA-based methods. Specifically, compared to the best baseline, HiMoLE achieves improvements of up to 3.0% on ID datasets and 5.0% on OOD datasets. (2) Although the recently proposed HydraLoRA and MixLoRA suggest their ability under multi-task learning scenarios, the lack of adequate sentence-level message integration for routing creates a bottleneck under out-of-distribution data, even resulting in worse performance in knowledge-intensive domains. (3) The generalization improvements from HiMoLE demonstrate domain-dependent variability, showing more pronounced gains in knowledge-intensive domains, with the performance enhancement compared to the best baseline increasing from 0.6% to 5.0% on OOD datasets. (4) All MoPE-based methods still face limitations in enhancing generalization ability when operating under a fixed number of parameters and limited data.

### 4.4   Effectiveness of HiMoLE on OOD samples

**Hierarchical Experts**   To substantiate the superiority of our hierarchical expert design, we conduct load balancing evaluations on out-of-distribution datasets. Through the visualization of expert selection logits from a randomly sampled OOD instance (Fig. 4, see the complete result in Appendix E.5), we observe that MixLoRA exhibits consistently severe load imbalance across layers compared to HiMoLE, with this disparity intensifying at deeper layers- particularly in its final layer where 78.3% of tokens disproportionately select Expert 4 with probabilities exceeding 0.85. Quantitatively, we adopt the MaxVio$_{global}$ metric from [38] (see the definition in Appendix E.3), to evaluate MoPE-layer balancing. As displayed in Fig. 4, HiMoLE consistently improved the load balancing among experts, with particularly notable improvements in knowledge-intensive domains (20.6% reduction in biomedical entity recognition). These results indicates that HiMoLE's hierarchical architecture effectively reduces expert redundancy through structural sparsity constraints, thereby promoting experts specialization and balanced expert utilization.

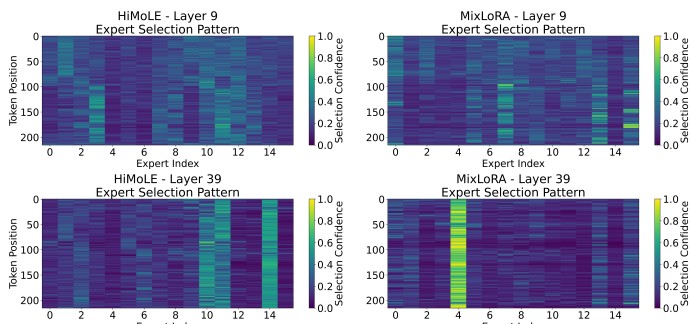
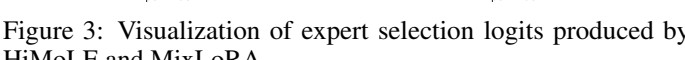

Figure 3: Visualization of expert selection logits produced by HiMoLE and MixLoRA.

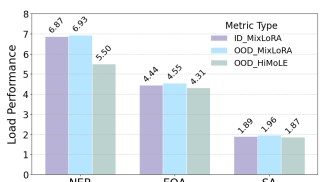

Figure 4: Comparison of experts load balance for different mixture of LoRA experts methods. We utilize MaxVio to evaluate the load performance.

Table 2: Robustness on character-level adversarial attack. Reported results are accuracy scores. The Robustness Ratio is defined as adversarial sample accuracy divided by original accuracy.

| Metric | Base | HiMoLE | MixLoRA |
|---|---|---|---|
| original | 0.488 | **0.733** | 0.695 |
| attacked | 0.196 | **0.330** | 0.280 |
| Robustness Ratio | 40.2% | **45.1%** | 40.3% |

Table 3: Ablation study on the two stage training strategy and diverse loss. Reported results are F1 scores. Please refer to Appendix E.5 for the complete results.

| Dataset | ID1 | ID2 | ID3 | ID4 | OOD |
|---|---|---|---|---|---|
| HiMoLE | 87.6 | 64.0 | 75.3 | 79.6 | 65.3 |
| -w/o. Two-stage Training | 77.7 | 32.1 | 64.1 | 60.8 | 53.5 |
| -w/o. Diverse Loss | 87.0 | 63.1 | 74.7 | 78.2 | 65.0 |

**Hierarchical Routing Strategy**  To further investigate the robustness of the hierarchical routing strategy, we compared its performance against the token routing method (MixLoRA) in the sentiment analysis task. This assessment involved generating adversarial out-of-distribution samples using the TextBugger [39] tool. Specifically, we randomly sampled 500 instances from the in-distribution test set and injected token-level noise via character-level perturbations.

As shown in Table 2, both routing strategies experienced significant performance degradation under adversarial attacks; however, hierarchical router demonstrated not only a superior absolute performance over token-level router, with a 6.0% improvement, but also exhibited a smaller decline in performance, with a 4.8% improvement over the token routing method. This indicates that hierarchical router possesses markedly stronger robustness against adversarial samples when compared to the base model and the token-level MoPE. The results further validate that HiMoLE, through its integration of global features, maintains stable performance against local perturbations by selecting appropriate experts via sentence-level semantic analysis, even in adversarial scenarios.

## 4.5 Ablations

**Two-stage Training Strategy**  We investigated the influence of different training strategies on model performance by conducting the experiment on the NER task. As evidenced in Table 3, when using single-stage joint training without preliminary KCG initialization, we observed a dramatic performance degradation of up to 31.9% F1 on the ID2 dataset. Furthermore, we examined the bad cases and found that the proportion of incorrect formats had increased significantly (see Appendix A). These observations indicate that (1) unstable optimization trajectories were adopted when learning router-expert interactions from random initialization, and (2) the absence of first-stage specialization prevents KCGs from developing domain-specific inductive biases, resulting in ambiguous routing signals. Thus, proper KCG initialization is critical for our hierarchical routing mechanism to function.

**Diversity Loss**  We investigated the importance of diverse loss on model performance. As evidenced in Table 3, omitting the diversity loss component leads to statistically significant performance degradation across both ID and OOD data. This consistent pattern reveals that diversity regularization can further mitigate expert group redundancy and improve model robustness.

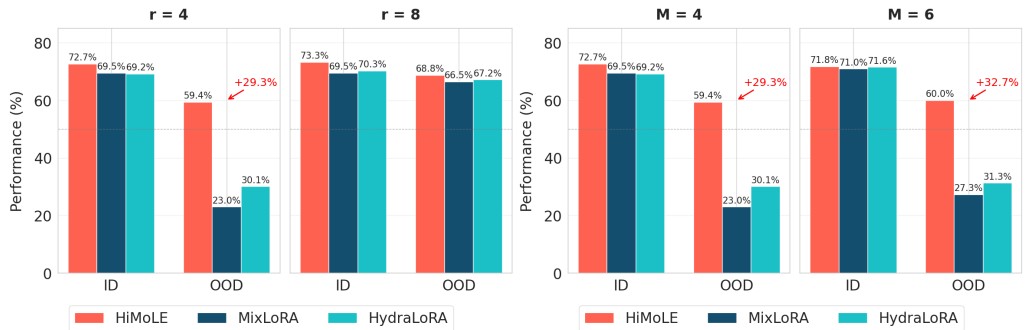

Figure 5: Hyper-parameter Analysis on LoRA rank $r$ and Knowledge Collaboration Experts Numbers $M$. Performance on OOD dataset are evaluated using REM (Appendix E.3).

## 4.6 Hyper-parameter Analysis

As illustrated in Figure 5, we further conduct hyper-parameter analysis on the sentiment analysis task examine the impact of the LoRA rank $r$ and the number of knowledge collaboration experts $M$. In experiments, we fix $r = 4$ when analyzing $M$, and conversely maintain $M = 4$ when investigating $r$. The results reveal that HiMoLE demonstrates superior robustness against variations in both parameters, consistently outperforming baseline methods across all configurations. Notably, while competing approaches exhibit significant performance degradation on OOD data with reduced LoRA ranks, HiMoLE achieves a remarkable OOD accuracy improvement from 31.3% to 60.0%, conclusively validating our method's effectiveness.

## 5 Conclusion

While PEFT methods like LoRA have significantly lowered the barriers for adapting large language models to downstream tasks, our investigation exposes their critical vulnerability to distributional shifts—particularly in knowledge-intensive domains. Our proposed HiMoLE framework alleviates this fundamental problem with by integrating hierarchical experts with a hierarchical routing strategy. This approach leverages sentence-level information to coarsely allocate experts to relevant subdomains and then refines the routing weights using token-level information, enabling efficient acquisition of new knowledge while preserving existing knowledge. By theoretically and empirically validating this approach across three representative NLP tasks, we establish a new paradigm for developing adaptable language models that achieve parameter efficiency with enhanced generalization capacity.

## Acknowledgement

This work is funded by Zhejiang Provincial "Jianbing" "Lingyan" Research and Development Program of China (2025C01129), National Natural Science Foundation of China (62302433, 62301480, U23A20496), and Hangzhou West Lake Pearl Project Leading Innovative Youth Team Project (TD2023017). This work was supported by Ant Group and Zhejiang University - Ant Group Joint Laboratory of Knowledge Graph.

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

# A Bad case studies

**Incorrect Format**   In the training ID data, the presence of numerous duplicate entity objects in the labels causes LoRA-based fine-tuning methods to inadvertently replicate this pattern. Consequently, when applied to OOD data, the model tends to generate more entities than necessary, often resulting in repetitive output behavior. This ultimately hinders its ability to produce correctly formatted JSON outputs as instructed. As illustrated in Table 4, this issue is particularly pronounced with both simple LoRA and token-level MoLE methods. In contrast, sentence-level MoLE and HiMoLE effectively address this problem. Notably, sentence-level MoLE ensures that all examples produce correctly formatted JSON outputs.

Table 4: Comparison of the Format Robustness

| Method | Base | LoRA | Token-MoLE | Sentence-MoLE | HiMoLE | HiMoLE - two stage training |
|---|---|---|---|---|---|---|
| Incorrect Format Ratio(%) | 1.19 | 8.33 | 7.14 | 0.00 | 2.38 | 44.0 |

**Misclassifications**   Analyzing the erroneous results revealed that models trained on ID data frequently mislabel certain symptoms as diseases when assessed on OOD data, failing to apply the general knowledge that differentiates the two. HiMoLE significantly mitigates this issue. Figure 6 showcases several comparative results between MixLoRA and HiMoLE.

# B Definitions of the Symbols

Table 5: Definitions of the Symbols used in the paper

| Symbol | Description |
|---|---|
| $i, m, n, l$ | the index number |
| $B, A$ | the low-rank matrices |
| $E$ | the weight matrix of a LoRA expert |
| $r, d_{\text{in}}, d_{\text{out}}, d_{\text{emb}}$ | the LoRA rank, the dimension of the input and the output of FFN layer, the dimension of the sentence embedding used for dataset clustering |
| $N$ | the number of the Knowledge Competition Groups |
| $M$ | the number of the Knowledge Collaboration Experts in a Knowledge Competition Group |
| $G_{\text{token}}, G_{\text{sen}}, G_{\text{hie}}$ | the gating weights matrix derived from the sentence-level router, the token-level router and the hierarchical router, respectively |
| $W_{\text{token}}, W_{\text{sen}}$ | the learnable parameters of the sentence-level router and the token-level router |
| $f_{\text{token}}, f_{\text{sen}}$ | the representation of the sentence-level router and the token-level router |
| $o$ | the output of a FFN layer |
| $h_{\text{token}}$ | token-level hidden representation |
| $h_{\text{sen}}$ | sentence-level hidden representation |
| $e$ | the output of a Knowledge Competition Group |
| $S$ | the cosine similarity between a pair of Knowledge Competition Groups |
| $\mathcal{L}_{\text{task}}, \mathcal{L}_{\text{aux}}, \mathcal{L}_{\text{diverse}}$ | the representations of overall loss, the task loss, the auxiliary loss and the diverse loss |
| $\nabla_\theta \mathcal{L}(\cdot)$ | the gradient of the parameter $\theta$ with the loss function $\mathcal{L}$ |
| $SimGrad$ | the expected pairwise gradient similarity from all different tokens |
| $V(g)$ | the variance of gradient |
| $\mathbb{E}$ | the notation of the expectation |
| $\mathbb{R}$ | the real number space |

| Base | MixLoRA | HiMoLE |
|---|---|---|
| Alkaptonuria is a rare genetic metabolic disorder characterized by the accumulation of homogentisic acid in the body. Affected individuals lack enough functional levels of an enzyme required to breakdown homogentisic acid. Affected individuals may have dark urine or urine that turns black when exposed to air. However, this change may not occur for several hours after urination and often goes unnoticed. Aside from dark urine that is present from infancy, affected individuals generally do not develop symptoms (asymptomatic) during infancy or childhood and often remain unaware of their condition until adulthood. Affected individuals eventually develop ochronosis, which is the bluish-black discoloration of connective and other tissue within the body. Affected individuals may develop discoloration of the skin overlying cartilage within the body such as over part of the outer ear. In some cases, the whites of the eyes (sclera) may also become discolored. In adulthood, affected individuals also develop progressive arthritis of the spine and large joints. The HGD gene codes for the enzyme required for the breakdown of homogentisic acid. Mutations in the HGD gene cause alkaptonuria. Alkaptonuria affects males and females in equal numbers, although symptoms tend to develop sooner and become more severe in males. More than 1,000 affected individuals have been reported in the medical literature. The exact incidence of alkaptonuria is unknown. In the United States it is estimated to occur in 1 in 250,000-1,000,000 live births. Alkaptonuria has been reported in all ethnic groups. Areas with increased frequencies of the disorder have been identified in Slovakia, the Dominican Republic and Germany. | Alkaptonuria is a rare genetic metabolic disorder characterized by the accumulation of homogentisic acid in the body. Affected individuals lack enough functional levels of an enzyme required to breakdown homogentisic acid. Affected individuals may have dark urine or urine that turns black when exposed to air. However, this change may not occur for several hours after urination and often goes unnoticed. Aside from dark urine that is present from infancy, affected individuals generally do not develop symptoms (asymptomatic) during infancy or childhood and often remain unaware of their condition until adulthood. Affected individuals eventually develop ochronosis, which is the bluish-black discoloration of connective and other tissue within the body. Affected individuals may develop discoloration of the skin overlying cartilage within the body such as over part of the outer ear. In some cases, the whites of the eyes (sclera) may also become discolored. In adulthood, affected individuals also develop progressive arthritis of the spine and large joints. The HGD gene codes for the enzyme required for the breakdown of homogentisic acid. Mutations in the HGD gene cause alkaptonuria. Alkaptonuria affects males and females in equal numbers, although symptoms tend to develop sooner and become more severe in males. More than 1,000 affected individuals have been reported in the medical literature. The exact incidence of alkaptonuria is unknown. In the United States it is estimated to occur in 1 in 250,000-1,000,000 live births. Alkaptonuria has been reported in all ethnic groups. Areas with increased frequencies of the disorder have been identified in Slovakia, the Dominican Republic and Germany. | Alkaptonuria is a rare genetic metabolic disorder characterized by the accumulation of homogentisic acid in the body. Affected individuals lack enough functional levels of an enzyme required to breakdown homogentisic acid. Affected individuals may have dark urine or urine that turns black when exposed to air. However, this change may not occur for several hours after urination and often goes unnoticed. Aside from dark urine that is present from infancy, affected individuals generally do not develop symptoms (asymptomatic) during infancy or childhood and often remain unaware of their condition until adulthood. Affected individuals eventually develop ochronosis, which is the bluish-black discoloration of connective and other tissue within the body. Affected individuals may develop discoloration of the skin overlying cartilage within the body such as over part of the outer ear. In some cases, the whites of the eyes (sclera) may also become discolored. In adulthood, affected individuals also develop progressive arthritis of the spine and large joints. The HGD gene codes for the enzyme required for the breakdown of homogentisic acid. Mutations in the HGD gene cause alkaptonuria. Alkaptonuria affects males and females in equal numbers, although symptoms tend to develop sooner and become more severe in males. More than 1,000 affected individuals have been reported in the medical literature. The exact incidence of alkaptonuria is unknown. In the United States it is estimated to occur in 1 in 250,000-1,000,000 live births. Alkaptonuria has been reported in all ethnic groups. Areas with increased frequencies of the disorder have been identified in Slovakia, the Dominican Republic and Germany. |

| Base | MixLoRA | HiMoLE |
|---|---|---|
| Fitz-Hugh-Curtis syndrome is a rare disorder that occurs almost exclusively in women. It is characterized by inflammation of the membrane lining the stomach (peritoneum) and the tissues surrounding the liver (perihepatitis). The muscle that separates the stomach and the chest (diaphragm), which plays an essential role in breathing, may also be affected. Common symptoms include severe pain in the upper right area (quadrant) of the abdomen, fever, chills, headaches, and a general feeling of poor health (malaise). Fitz-Hugh-Curtis syndrome is a complication of pelvic inflammatory disease (PID), a general term for infection of the upper genital tract in women. Infection is most often caused by Neisseria gonorrhoeae and Chlamydia trachomatis. A diagnosis of Fitz-Hugh-Curtis syndrome is made through the exclusion of other causes of upper right abdominal pain. A diagnosis may be confirmed with a variety of specialized tests including x-ray examination, diagnostic laparoscopy, and certain laboratory exams. X-ray examination may include ultrasound, chest or stomach radiographs, and computed tomography (CT) scanning. X-rays are used to rule out other possible conditions or reveal characteristic inflammation of the perihepatic region. During a laparoscopy, a small, thing tube is inserted in the abdominal cavity through a small incision in the stomach. A laparoscopic exam allows a physician to view the liver and surrounding tissue. Laboratory exams can identify infection with Chlamydia trachomatis or Neisseria gonorrhoeae. | Fitz-Hugh-Curtis syndrome is a rare disorder that occurs almost exclusively in women. It is characterized by inflammation of the membrane lining the stomach (peritoneum) and the tissues surrounding the liver (perihepatitis). The muscle that separates the stomach and the chest (diaphragm), which plays an essential role in breathing, may also be affected. Common symptoms include severe pain in the upper right area (quadrant) of the abdomen, fever, chills, headaches, and a general feeling of poor health (malaise). Fitz-Hugh-Curtis syndrome is a complication of pelvic inflammatory disease (PID), a general term for infection of the upper genital tract in women. Infection is most often caused by Neisseria gonorrhoeae and Chlamydia trachomatis. A diagnosis of Fitz-Hugh-Curtis syndrome is made through the exclusion of other causes of upper right abdominal pain. A diagnosis may be confirmed with a variety of specialized tests including x-ray examination, diagnostic laparoscopy, and certain laboratory exams. X-ray examination may include ultrasound, chest or stomach radiographs, and computed tomography (CT) scanning. X-rays are used to rule out other possible conditions or reveal characteristic inflammation of the perihepatic region. During a laparoscopy, a small, thing tube is inserted in the abdominal cavity through a small incision in the stomach. A laparoscopic exam allows a physician to view the liver and surrounding tissue. Laboratory exams can identify infection with Chlamydia trachomatis or Neisseria gonorrhoeae. | Fitz-Hugh-Curtis syndrome is a rare disorder that occurs almost exclusively in women. It is characterized by inflammation of the membrane lining the stomach (peritoneum) and the tissues surrounding the liver (perihepatitis). The muscle that separates the stomach and the chest (diaphragm), which plays an essential role in breathing, may also be affected. Common symptoms include severe pain in the upper right area (quadrant) of the abdomen, fever, chills, headaches, and a general feeling of poor health (malaise). Fitz-Hugh-Curtis syndrome is a complication of pelvic inflammatory disease (PID), a general term for infection of the upper genital tract in women. Infection is most often caused by Neisseria gonorrhoeae and Chlamydia trachomatis. A diagnosis of Fitz-Hugh-Curtis syndrome is made through the exclusion of other causes of upper right abdominal pain. A diagnosis may be confirmed with a variety of specialized tests including x-ray examination, diagnostic laparoscopy, and certain laboratory exams. X-ray examination may include ultrasound, chest or stomach radiographs, and computed tomography (CT) scanning. X-rays are used to rule out other possible conditions or reveal characteristic inflammation of the perihepatic region. During a laparoscopy, a small, thing tube is inserted in the abdominal cavity through a small incision in the stomach. A laparoscopic exam allows a physician to view the liver and surrounding tissue. Laboratory exams can identify infection with Chlamydia trachomatis or Neisseria gonorrhoeae. |

| Base | MixLoRA | HiMoLE |
|---|---|---|
| Spondyloepiphyseal dysplasia tarda (SEDT; SEDL) is a rare, hereditary skeletal disorder that only affects males. Physical characteristics include moderate short stature (dwarfism), moderate-to-severe spinal deformities, barrel-shaped chest, disproportionately short trunk, and premature osteoarthritis. SEDT does not exhibit any ethnic predisposition. Affected individuals have been described in European, American, Asian, and Australian populations (but not in African-Americans to date). One estimate suggests that the incidence is 2 persons per million. | Spondyloepiphyseal dysplasia tarda (SEDT; SEDL) is a rare, hereditary skeletal disorder that only affects males. Physical characteristics include moderate short stature (dwarfism), moderate-to-severe spinal deformities, barrel-shaped chest, disproportionately short trunk, and premature osteoarthritis. SEDT does not exhibit any ethnic predisposition. Affected individuals have been described in European, American, Asian, and Australian populations (but not in African-Americans to date). One estimate suggests that the incidence is 2 persons per million. | Spondyloepiphyseal dysplasia tarda (SEDT; SEDL) is a rare, hereditary skeletal disorder that only affects males. Physical characteristics include moderate short stature (dwarfism), moderate-to-severe spinal deformities, barrel-shaped chest, disproportionately short trunk, and premature osteoarthritis. SEDT does not exhibit any ethnic predisposition. Affected individuals have been described in European, American, Asian, and Australian populations (but not in African-Americans to date). One estimate suggests that the incidence is 2 persons per million. |

Figure 6: Misclassifications Cases. Text highlighted in green represents correct entity annotations, while yellow represents incorrect entity annotations.

## C Auxiliary Loss

Given $N$ experts indexed by $i = 1$ to $N$ and a batch $B$ with $T$ tokens. Let $G(\cdot)$ denotes the top-k router, $F_i$ is the fraction of tokens dispatched to expert, and $P_i$ i is the fraction of the router probability allocated for expert $i$. The final loss is then multiplied by the expert count N to keep the loss constant as the number of experts varies, which can be formulated as following:

$$F_i = \frac{1}{T} \sum_{x \in \mathcal{B}} \mathbb{I}\left(\text{argmax}_k R(x)_k = i\right), P_i = \frac{1}{T} \sum_{x \in \mathcal{B}} R(x_i), \tag{12}$$

$$\mathcal{L}_{\text{aux}} = N \cdot \sum_{i=1}^{N} F_i \cdot P_i, \tag{13}$$

## D Proof

Here we provide the proof of Theorem 1.

**Lemma 2** *Let $t$ represent distinct tokens and $s$ represent sentences. To simplify, let's assume that the sample gradients are a set of independent and identically distributed unit vectors. In this case, $SimGrad$ can be expressed as $||\mathbb{E}(\nabla_{\theta_i} L(h_t))||^2$:*

$$\begin{aligned} SimGrad &= \mathbb{E}(cos(g_t, g_{t'})) = \mathbb{E}(g_t \cdot g_{t'}) \\ &= \mathbb{E}_{g_t}(\mathbb{E}_{g'_t}[g_t \cdot g_{t'}|g_t]) = \mathbb{E}_{g_t}(g_t \cdot \mathbb{E}_{g_{t'}}(g_{t'})) = \mathbb{E}_{g_t}(g_t) \cdot \mathbb{E}_{g_{t'}}(g_{t'}) \\ &= ||\mathbb{E}_{g_t}||^2 = ||\mathbb{E}(\nabla_{\theta_i} L(x_t))||^2 \end{aligned} \tag{14}$$

**Proof D.1** *For the $i$-th expert $E_i$, let $A_{s,t}$ denotes $\mathbb{I}\left(E_i \in argmax_k(f_{sen}(h_s) \odot f_{token}(h_t))\right)$, $B_{s,t}$ denotes $\mathbb{I}(E_i \in argmax_k(f_{token}(h_t)))$. For hierarchical router $G_{hie}$, and token router $G_{token}$, their gradient operators can be written as follows:*

$$\begin{aligned} \nabla_{\theta_i} \mathcal{L}_{hie} &= \mathbb{I}\left(E_i \in argmax_k(G_{hie})\right) \cdot \nabla\theta_i \mathcal{L}(h_{s,t}) \\ &= \mathbb{I}\left(E_i \in argmax_k(f_{sen}(h_s) \odot f_{token}(h_t))\right) \cdot \nabla\theta_i \mathcal{L}(h_{s,t}) \\ &= A_{s,t} \cdot \nabla\theta_i \mathcal{L}(h_{s,t}) \end{aligned} \tag{15}$$

$$\begin{aligned} \nabla_{\theta_i} \mathcal{L}_{token} &= \mathbb{I}\left(E_i \in argmax_k(G_{token})\right) \cdot \nabla\theta_i \mathcal{L}(x_{s,t}) \\ &= \mathbb{I}\left(E_i \in argmax_k(f_{token}(x_t))\right) \cdot \nabla\theta_i \mathcal{L}(h_{s,t}) \\ &= B_{s,t} \cdot \nabla\theta_i \mathcal{L}(h_{s,t}) \end{aligned} \tag{16}$$

*Combining Lemma 2, $\Delta_{\theta_i}$ can be written as follows:*

$$\begin{aligned} \Delta_{\theta_i} &= SimGrad_{hie} - SimGrad_{token} \\ &= ||\mathbb{E}(\nabla_{\theta_i} L_{hie})||^2 - ||\mathbb{E}(\nabla_{\theta_i} L_{token})||^2 \\ &= [\mathbb{E}(A_{s,t})^2 - \mathbb{E}(B_{s,t})^2] \cdot \mathbb{E}(\nabla\theta_i L(h_{s,t}))^2 \end{aligned} \tag{17}$$

*Since $A_{s,t}, B_{s,t} \in \{0, 1\}$, we have:*

$$\mathbb{E}(A_{s,t})^2 = \mathbb{E}(A_{s,t}), \mathbb{E}(\mathbb{B}_{s,t})^2 = \mathbb{E}(B_{s,t})$$

*Next, we analyze the relationship between $A_{s,t}$ and $B_{s,t}$. The hierarchical router ensures that the effective selection of token router is not overlooked by positively adjusting the scores. Additionally, by introducing global information through sentence features, it uncovers experts that are neglected under local token features, which guarantee that:*

$$B_{s,t} = 1 \Rightarrow A_{s,t} = 1, \text{ but not vice versa, } A_{s,t} = 1 \not\Rightarrow B_{s,t} = 1$$

*As a result,*

$$\begin{aligned} \Delta_{\theta_i} &= [\mathbb{E}(A_{s,t})^2 - \mathbb{E}(B_{s,t})^2] \cdot \mathbb{E}(\nabla\theta_i L(h_{s,t}))^2 \\ &= [\mathbb{E}(A_{s,t}) - \mathbb{E}(B_{s,t})] \cdot \mathbb{E}(\nabla\theta_i L(h_{s,t}))^2 >= 0 \end{aligned} \tag{18}$$

Table 6: Description of Datasets used in experiments.

| Task | Domain | Train | Test ID | OOD |
|------|--------|-------|---------|-----|
| NER | Biomedical | 35132 | 2060/1267/2764/2746 | 684 |
| SA | Social Science | 11257 | 2374 | 1465 |
| EQA | General | 87599 | 10570 | 3882 |

Now we turn to the proof of Theorem 2.

**Lemma 3** *Let $g$ denotes the gradient, $\mathbb{E}(g)$ denote the average gradient and $m$ denote the number of tokens, then $\mathbb{E}(g)^2$ can be written as follows:*

$$\mathbb{E}(g)^2 = \frac{1}{m^2} \left( \sum_t \|g_t\|^2 + 2 \sum_{t \neq t'} g_t \cdot g_{t'} \right) \tag{19}$$

**Lemma 4** *Let $V(g)$ represents the gradient variance, then $V(g) \propto -SimGrad$:*

$$
\begin{aligned}
V(g) &= \mathbb{E}\left[ \|(g_t - \mathbb{E}(g_t)\|^2 \right] = \frac{1}{m} \sum_t \|g_t - \mathbb{E}(g)\|^2 \\
&= \frac{1}{m} \left( \|g_t\|^2 - 2g_t \cdot \mathbb{E}(g) + \mathbb{E}(g)^2 \right) \\
&= \frac{1}{m} \sum_t \|g_t\|^2 - \mathbb{E}(g)^2
\end{aligned}
\tag{20}
$$

*Combing with Lemma 3, $V(g)$ can be written as:*

$$V(g) = \frac{m-1}{m^2} \sum_t \|g_t\|^2 - \frac{2}{m^2} \sum_{t \neq t'} g_t \cdot g_{t'} \propto -SimGrad \tag{21}$$

**Proof D.2** *By Theorem 1, $SimGrad_{hie} >= SimGrad_{token}$. By lemma 4, higher $SimGrad$ indicates lower gradient variance, hence $V(g_{hie}) <= V(g_{token})$.*

# E Experiments Details

## E.1 Datasets

Table 6 summarizes the datasets used in our experiments, including their task names, respective domains, the number of training and test sets. For Biomedical NER, the ID test data was partitioned into 4 sub-datasets using feature-based $K$-means clustering. All datasets are downloaded from HuggingFace using the DATASETS library in Python. Additionally, we provide a UMAP visualization of the datasets in Fig. 7.

### E.2 Hyperparameters and Implementation Details

Table 7: Hyperparameter configurations of LoRA, MixLoRA/HydraLoRA and HiMoLE for fine-tuning LLaMA2-7B and OneKE-13B.

| Metric | LoRA | MixLoRA/HydraLoRA | HiMoLE |
|---|---|---|---|
| Cutoff Length | 1024 | 1024 | 1024 |
| Learning Rate | 3e-4 | 3e-4 | 3e-4(stage1), 3e-5(stage2) |
| Optimizer | AdamW | AdamW | AdamW |
| Batch size | 16 | 16 | 16 |
| Dropout | 0.05 | 0.05 | 0.05 |
| Where | Up, Down, Gate | Up, Down, Gate | Up, Down, Gate |
| LoRA Rank | 80 | 8 | 8 |
| LoRA Alpha | 160 | 16 | 16 |
| Top-K | - | 2 | 2 |

We set a maximum of 10,000 training steps and perform evaluations on the validation sets of all benchmarks every 50 steps. If there is no improvement on the validation set for 10 consecutive evaluations, we will terminate the training early. The best checkpoint, identified by the highest average accuracy across all benchmarks, is then selected for evaluation on the test set.

All experiments are conducted with GPUs having 24GB memory (RTX 4090) for 7B models, GPUs having 40GB memory (RTX A100) for 13B models, and setup with Python 3.8 and Ubuntu 22.04 on x86-64 CPUs.

### E.3 Evaluation Metrics

**Performance** For the metrics in Table 1: In NER, F1 stands for the average F1 score, P stands for the average precision, R stands for the average recall; In SA, EM stands for the exact match, REM stands for the relaxed exact match( we treat the 'positive' label as synonymous with 'optimism' and the 'negative' label as synonymous with 'pessimism'); In QA, EM stands for the exact match, ROUGE-2 is employed to captures phrases that hold vital context.

**Load Balance** MaxVio is used to quantify the degree of load balance of an MoE layer, defined as $\text{MaxVio} = \frac{\max_i \text{Load}_i - \overline{\text{Load}_i}}{\overline{\text{Load}_i}}$,, where $\text{Load}_i$ represents the number of tokens assigned to the $i$-th expert, and $\overline{\text{Load}_i}$ denotes the expected expert load under perfect load balance. In $\text{MaxVio}_{\text{global}}$, $\text{Load}_i$ is calculated on the whole validation set.

### E.4 Additional Experiments and Analysis

**Impact of Two-stage Training Strategy** To disentangle the effects of initialization from our hierarchical architecture, we add baseline comparisons where MixLoRA and HydraLoRA are trained with the same Stage 1 initialization as HiMoLE on NER task. As shown in Table 8, although other methods demonstrate improvements, the two-stage training approach combined with the hierarchical mixture of LoRA experts still delivers the best performance across both in-distribution and out-of-distribution settings, with essential improvement under OOD setting.

Table 8: Impact of Training Strategy across mixture of LoRA experts methods. The reported results(%) are F1.

| Training Strategy | HiMoLE | | MixLoRA | | HydraLoRA | |
|---|---|---|---|---|---|---|
| | ID | OOD | ID | OOD | ID | OOD |
| two stage | **77.9** | **65.3** | 76.4 | 63.0 | 77.2 | 77.3 |
| one stage | 61.7 | 53.5 | 76.0 | 61.0 | 77.3 | 62.9 |

**Inference Latency**   HiMoLE adds lightweight gating networks, maintaining nearly the same parameter count as MixLoRA. For inference, the speed is primarily influenced by the base model. Since the parameters of PEFT modules constitute a small fraction of the total model parameters (ranging from 0.65% to 1.05% as shown in Table 9), the inference latency differences across are minimal. Table 9 presents the latency and parameter count during inference using Llama2-7B with different Mixture of LoRA Experts methods, evaluated on the OPTIMISM dataset using a single RTX 4090 GPU. Latency was recorded over 50 random samples. The results show nearly equal latency, but HiMoLE exhibits the highest model performance.

### E.5   Complete Results

Table 10 presents the complete results of the comparative performance for the NER task in Table 1. Fig. 8 presents the complete results of the routing logits discussed in Section 4.3. Table 11 displays the complete results of the ablation experiments in Table 3.

## F   Training Strategy

---

**Algorithm 1:** HiMoLE Two-Stage Training

---

**Input:** LLM's frozen weights $W_0$, training data $\mathcal{D}$ (composed of $(s, y)$ pairs), pre-trained encoder Encoder$(\cdot)$, cluster count $N$
**Output:** Optimized experts and routers

**Stage** *1: Knowledge Competition Group Initialization*

**foreach** *sentence $s_i \in \mathcal{D}$* **do**
   emb$_i \leftarrow$ Encoder$(s_i)$ ;                           `//Generate semantic embeddings`
$\{\mathcal{C}_1, ..., \mathcal{C}_N\} \leftarrow K\text{-means}(\{\text{emb}_i\}, N)$ ;         `//Cluster embeddings`
**For** $k \leftarrow 1$ **to** $N$ **in parallel**
   $\mathcal{D}_k \leftarrow \{s_i, y_i)| \text{emb}_i \in \mathcal{C}_k\}$ ;               `//Build sub-dataset`
   $\theta_{\text{KCG}_k} \leftarrow \arg\min_\theta \sum_{(s,y) \in \mathcal{D}_k} \mathcal{L}_{\text{task}}(f_\theta(s), y)$ ;   `//Train KCGs in parallel`

**Stage** *2: Joint Optimization of Experts & Routers*

Initialize routing parameters $\phi$, load pre-trained $\{\theta_{\text{KCE}_i}\}_{i=1}^{N \times M}$
**while** *not converged* **do**
   **for** *batch $\mathcal{B} \subset \mathcal{D}$* **do**
      **foreach** $(s, y) \in \mathcal{B}$ **do**
         $g \leftarrow$ Router$_\phi(s)$ ;                  `//Routing weights`
         $\hat{y} \leftarrow W_0 s + \sum_{i=1}^{N \times M} g_i f_{\theta_i}(s)$ ;     `//Weighted combination`
      Compute $\mathcal{L}_{\text{task}} = \ell(\hat{y}, y), \mathcal{L}_{\text{aux}}, \mathcal{L}_{\text{diverse}}$ ;     `//Compute loss`
      Update $\theta, \phi \leftarrow \theta, \phi - \eta \nabla(\mathcal{L}_{\text{task}} + \alpha \mathcal{L}_{\text{aux}} + \beta \mathcal{L}_{\text{diverse}})$

---

## G   Limitations and Future work

In this section, we discuss the potential limitations of our proposed method HiMoLE. Firstly, has shown effectiveness in addressing simple OOD scenarios, it still struggles to deal with hard OOD samples(e.g., in biomedical NER, it fails to outperform the base model on the OOD dataset). Future

Table 9: Inference Latency vs Performance across mixture of LoRA experts methods. The reported performance is evaluated using REM.

|  | HiMoLE | MixLoRA | HydraLoRA |
|---|---|---|---|
| %Param | 1.05 | 1.05 | 0.65 |
| Latency(s) | 127 | 119 | 122.0 |
| Performance(%) | **68.8** | 66.5 | 67.2 |

Table 10: Complete NER Results

| Dataset | ID1 | | | ID2 | | | ID3 | | | ID4 | | | OOD | | |
|---|---|---|---|---|---|---|---|---|---|---|---|---|---|---|---|
| Metric | F1 | P | R | F1 | P | R | F1 | P | R | F1 | P | R | F1 | P | R |
| Base Model | 59.1 | 60.6 | 59.0 | 42.4 | 52.0 | 40.4 | 51.3 | 53.4 | 51.8 | 52.3 | 57.5 | 51.4 | 65.8 | 63.1 | 76.4 |
| LoRA | 85.9 | 86.7 | 86.0 | 61.4 | 68.8 | 60.2 | 78.2 | 80.0 | 78.1 | 66.2 | 72.1 | 65.0 | 61.9 | 56.0 | 77.8 |
| MixLoRA | 87.4 | 88.4 | 87.3 | 60.6 | 64.2 | 59.9 | 73.4 | 75.8 | 73.0 | 77.3 | 79.5 | 77.0 | 61.0 | 59.1 | 73.7 |
| HydraLoRA | 87.8 | 88.4 | 87.8 | 63.2 | 65.2 | 62.3 | 75.1 | 75.9 | 74.4 | 78.2 | 79.4 | 77.2 | 62.9 | 59.4 | 75.5 |
| HiMoLE | 87.6 | 88.1 | 87.8 | 64.0 | 67.8 | 63.5 | 75.3 | 75.8 | 73.4 | 79.6 | 79.5 | 79.8 | 65.3 | 64.4 | 74.3 |

Table 11: Detailed Ablation Results

| Dataset | ID1 | | | ID2 | | | ID3 | | | ID4 | | | OOD | | |
|---|---|---|---|---|---|---|---|---|---|---|---|---|---|---|---|
| Metric | F1 | P | R | F1 | P | R | F1 | P | R | F1 | P | R | F1 | P | R |
| HiMoLE | 87.6 | 88.1 | 87.8 | 64.0 | 67.8 | 63.5 | 75.3 | 75.8 | 73.4 | 79.6 | 79.5 | 79.8 | 65.3 | 64.4 | 74.3 |
| -two stage training | 77.7 | 80.5 | 77.0 | 32.1 | 45.2 | 30.5 | 54.3 | 62.0 | 53.2 | 60.8 | 66.0 | 60.1 | 53.5 | 57.7 | 59.0 |
| -diverse loss | 87.0 | 87.5 | 87.3 | 62.7 | 66.2 | 62.4 | 74.9 | 77.2 | 74.8 | 78.2 | 79.8 | 78.4 | 65.0 | 62.8 | 76.7 |

work will incorporate data augmentation to handle with hard OOD samples. Secondly, we constrain the model size to 13B and limit the number of LoRA experts to 16 due to resource and time limitations. As expert numbers scale, the interaction mode among experts would be more intricate and more sophisticated routing topologies like graph can be introduced to adapt to the nuanced patterns that emerge. Subsequent research be will conducted on the larger LLMs and more LoRA experts with more complex interaction mechanisms. This dual-axis expansion (model size + adaptive expert management) could unlock new robustness frontiers without proportional computational overhead.

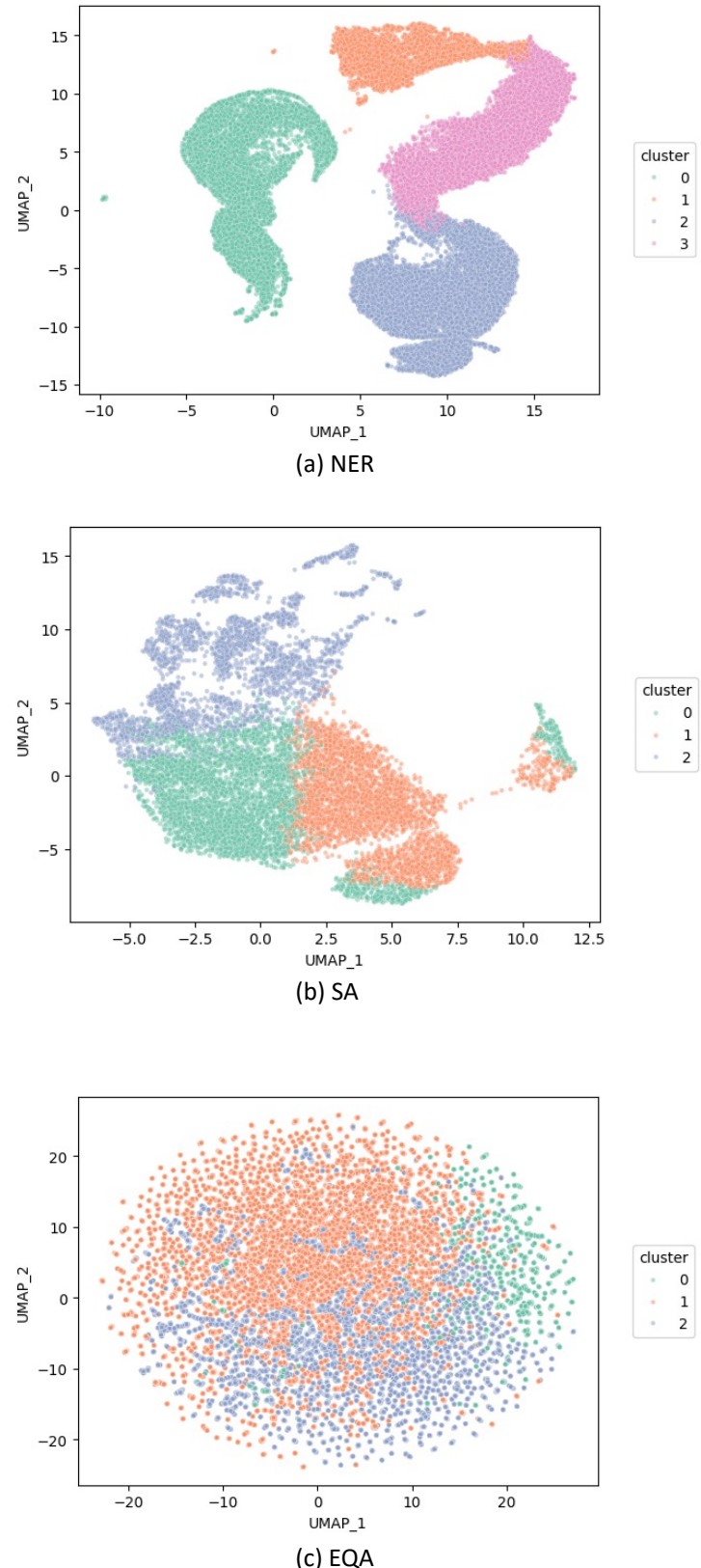

Figure 7: UMAP visualization of the datasets. Different colors correspond to different sub-datasets for KCGs initialization.

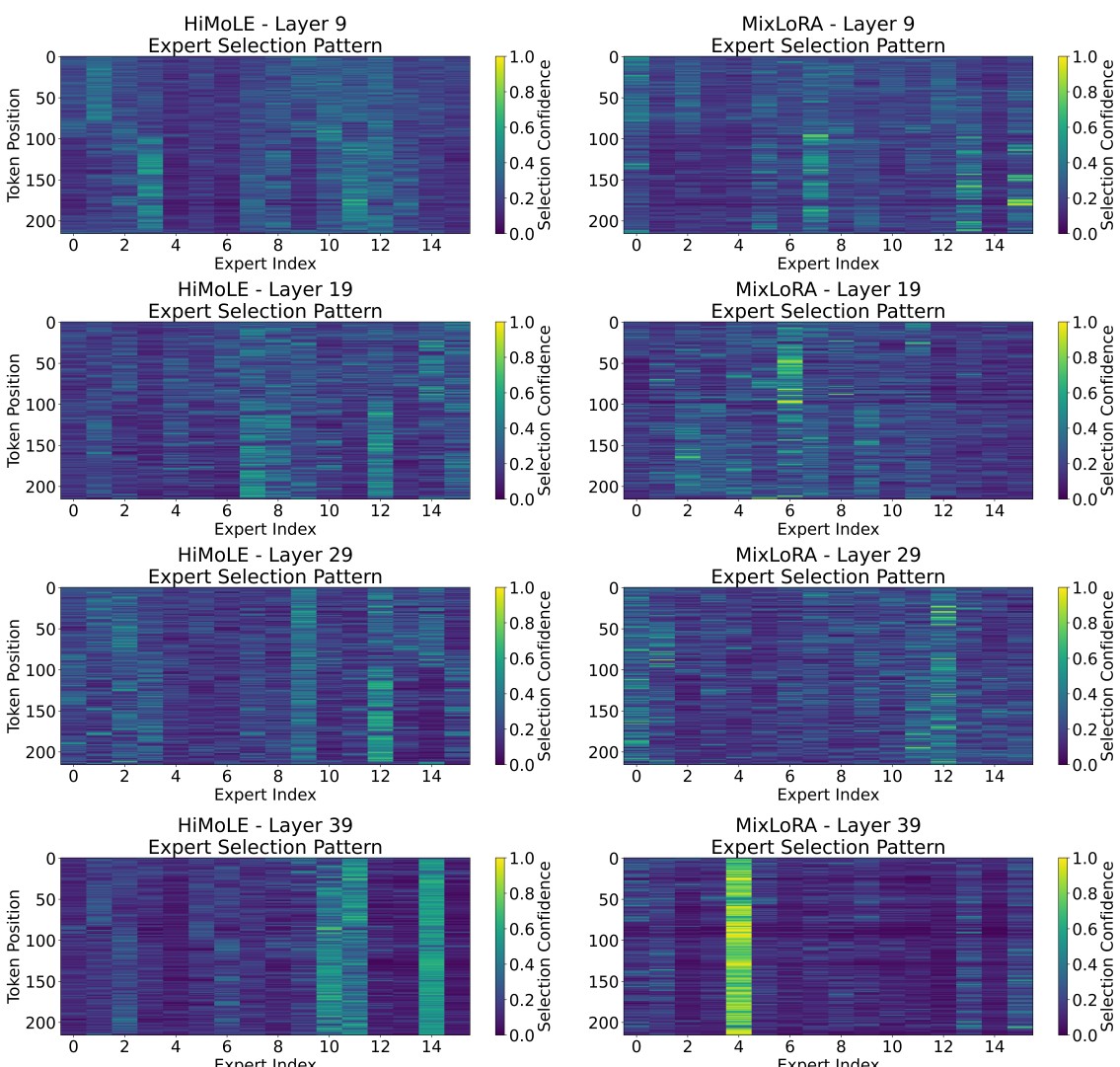

Figure 8: Expert Logits selection pattern.

