# OpenReview forum: "HiMoLE: Towards OOD-Robust LoRA via Hierarchical Mixture of Experts"
_NeurIPS.cc/2025/Conference — NeurIPS 2025 poster_

### Official Review · Reviewer_Q7d6 · 2025-07-01

**Clarity:** 2
**Significance:** 3
**Originality:** 3
**Rating:** 4
**Confidence:** 3

**Summary:**

To address the generative power and robustness of traditional LoRA in OOD scenarios, this paper proposes HiMoLE, an optimized LoRA architecture that integrates hierarchical expert structures (Knowledge Competition Groups (KCGs) and Knowledge Collaboration Experts (KCEs)) with a two-tiered routing mechanism (sentence-level + token-level router). It consists of a two-stage training and diversity loss design, and is empirically evaluated to outperform existing methods on multiple NLP tasks.

**Questions:**

1. HiMoLE design is complicated, but the depth of ablation study is not deep enough. E.g., is the co-existence of “sentence-level + token-level” routing necessary? Does “cross-group collaboration” between KCEs lead to performance improvements? Is it better to replace diversity loss with negatively correlated regular terms?
2. During the training process, the experts are fixed first and then tuned later: if there is a bias in the initial KCG, will the subsequent global tuning stage fall into local optimization?

**Ethical Concerns:**

["NO or VERY MINOR ethics concerns only"]

**Final Justification:**

Thanks for your detailed reply. At this stage I think my concerns are mostly addressed.

**Limitations:**

Yes.

**Paper Formatting Concerns:**

N/A.

**Quality:**

2

**Strengths And Weaknesses:**

Pros:
1. Robustness of LoRA under OOD (out-of-distribution) conditions is an important research issue.
2. HiMoLE introduces a two-tier hierarchical structure, more fine-grained expert design, and hierarchical routing to enable the model to better handle semantic heterogeneity.
3. This paper provides theoretical analysis and experimental evaluation to show the performance of HiMoLE under ID and OOD conditions.

Cons:
1. Missing overhead analysis: HiMoLE introduces hierarchical expert structure + hierarchical routing + two-stage training, which introduces large amount of parameter routing computation, expert switching overhead and pre-clustering tuning.
2. Lacking sensitivity to the number of clusters N and experts M, i.e., lack of sensitivity analysis for K-Means clustering.
3. The theoretical part neglects practical analysis and empirical proofs. E.g., Lemma 1 and Theorem 2 belong to the “generalized” SGD generalization framework without combining with the practical structure or the features of the loss function.
4. The paper piles up too much customized terminology.
5. For evaluation, the OOD of the NER, SA, and QA is still a controlled division within the domain, although it has been transformed into a distribution. More extreme OOD settings can be added, such as domain transfer (e.g., migrating from legal QA to medical QA), and practical user dataset.

---

> ### Author Rebuttal · Authors · 2025-07-31
>
> Thank you for your thoughtful feedback and constructive suggestions. Below, we address the concerns raised:
> - W1: Missing overhead analysis
>
>   We acknowledge the lack of explicit discussion on computational overhead and  analyze the costs introduced by HiMoLE’s hierarchical structure as below:
>   - Parameter Routing and Inference scalability: HiMoLE adds **lightweight gating networks, maintaining nearly the same parameter count as MixLoRA**. For inference, the speed is **primarily influenced by the base model**. Since the parameters of PEFT modules constitute a small fraction of the total model parameters (ranging from 0.65% to 1.05% as shown in the following Table), the inference latency differences across are minimal. The following presents the latency and parameter count during inference using Llama2-7B with different Mixture of LoRA Experts methods, evaluated on the OPTIMISM dataset using a single RTX 4090 GPU.  Latency was recorded over 50 random samples. The results show **nearly equal latency, but HiMole exhibits the highest model performance**.
>
>     |  | HiMole | MixLoRA | HydraLoRA |
>     | :--- | :--- | :--- | :--- |
>     | %Param | 1.05 | 1.05 | 0.65 |
>     | Latency(s) | 127 | 119 | 122 |
>     | Performance(%) | 68.8 | 66.5 | 67.2 |
>
>   - Two-Stage Training: The two-phase training incurs extra cost, but Stage 1 is amortized across downstream tasks. Stage 2 converges faster than end-to-end training due to better initialization. Meanwhile，the pre-clustering encoder's parameter count (e.g., 500M params) and FLOPs are orders of magnitude smaller than modern LLMs (e.g., 7B-13B params), **making clustering computationally trivial**. In our experiments, pre-clustering consumed < 2% of total training time (e.g., 0.2 GPU hours vs. 12 hours on NER task ). As validated in [1], **the two-stage training approach is more compute efficient** than training a larger generalist LLM.
>   We acknowledge HiMoLE is not zero-overhead, but **its costs are bounded**. Crucially, these overheads are **dwarfed by HiMoLE’s robustness gains under distribution shift**.
>
> - W2: Lack of sensitivity analysis
>
>   We acknowledge the lack of sensitivity experiments. The number of clusters (N) was determined using the silhouette coefficient describing the clustering quality during the pre-clustering stage.  As validated in the Section 4.5 of HydraLoRA[2], the number k of clusters is NOT a sensitive parameter with a wide range of reasonable number k of clusters performing decently well in all settings; K-means is simple but effective with the  sophisticated hyper-parameter search approaches.
>
>   We further evaluate HiMoLE with M=6 experts and the LoRA rank of 4, as shown in the table below. The results demonstrate that HiMoLE exhibits **stronger robustness to variations in the number of experts and LoRA rank**, while consistently achieving the best performance across all configurations. Notably, under the OOD setting, it delivers a substantial performance gain (31.3% → 60.0%), further validating the effectiveness of our method.
>
>   | Setting | Accuracy(%)| HiMoLE | MixLoRA | HydraLoRA |
>   | :--- |:--- | :--- | :--- | :--- |
>   | M=4 | ID | 72.7 | 69.5 | 69.2 |
>   |  | OOD | **59.4** | 23.0 | 30.1 |
>   | M=6 | ID | 71.8 | 71.0 | 71.6 |
>   |  | OOD | **60.0** | 27.3 | 31.3 |
> - W3: Lack of practical analysis and empirical proofs
>
>   We appreciate the your emphasis on practical alignment. While we employ a generalized SGD framework, we explicitly bridge this theory to HiMoLE’s structure through the following connection:
>   1. We acknowledge that the presentation in the manuscript focuses primarily on establishing the core theoretical generalization guarantees within a generalized SGD framework for our specific problem setting (Lemma 1, Theorem 2). We agree that explicitly incorporating the exact structural details of the loss function into these specific lemmas/theorems would further strengthen the practical connection. However, we respectfully clarify that the core theoretical mechanism governing generalization improvement identified in our analysis – **the reduction of generalization error driven by a decrease in the variance of the stochastic gradients** – is a fundamental and widely applicable principle in SGD analysis, largely independent of the specific loss function structure[3]. This principle justifies the applicability of the generalized framework in our work.
>   2. As shown in Table 3, in the ablation of diverse loss, where we maintain the same loss features, it still outperformers other MOPE methods on OOD dataset.
> - W4: Too much customized terminology
>
>   Thank you for highlighting this issue. We acknowledge that the initial draft overused specialized terminology, which may have hindered readability. To address this, we will simplify the language in the revised manuscript.
> - W5: Evaluation on extreme OOD data
>
>   Thank you for the suggestion. We further evaluate our model on a challenging out-of-distribution (OOD) dataset from a different domain—specifically, the Politics subset of CrossNER [4]. Results show that while our method still improves performance on these extreme OOD samples, the gains are more modest compared to OOD cases within the domain.
>
>   |  | Base Model | HiMole | MixLoRA | HydraLoRA |
>   | :--- | :--- | :--- | :--- | :--- |
>   | F1(%) | 61.5 | 66.9 | 65.8 | 66.5 |
>
> - Q1: Ablation study
>
>   As shown in Fig 1.d and illustrated in Section 3.1, there is **a key trade-off between routing granularities** : Token-level routing delivers top in-distribution accuracy by using fine-grained context, but it overfits and performs poorly out-of-distribution. Sentence-level routing is more robust out-of-distribution through its focus on overall meaning, yet it loses in-distribution accuracy by overlooking local details. Hence we integrate sentence-level routing and token-level routing in a hierarchical manner to enhance performance on both in-distribution (ID) and out-of-distribution (OOD) data.
>
>   To validate that cross-group collaboration among KCEs contributes to performance gains, we conduct an ablation study on the NER task. Specifically, we compare HiMoLE against the single KCG setting(which lacks cross-group collaboration) on the subdataset aligned with the KCG’s distribution. As shown in the following table, the substantial performance improvement confirms that cross-group collaboration enhances model effectiveness.
>
>   Regarding the loss function, the diverse loss **encourages expert groups to specialize in distinct task clusters**, thereby preventing redundancy and enhancing performance, which has been proved in Section 4.5. While negatively correlated regularization could theoretically achieve a similar effect, we found that **applying it to LLMs introduces computational overhead in practice** due to the cost of calculating parameter-wise negative correlations during training.
>
>   |  | ID1 | ID2 | ID3 | ID4 |
>   | :--- | :--- | :--- | :--- | :--- |
>   | KCG | 85.3 | 47.5 | 69.5 | 71.2 |
>   | HiMoLE | 87.6 | 64.0 | 75.3 | 79.6 |
>
> - Q2: Initialization
>
>   We appreciate the reviewer’s insightful question regarding the sensitivity of our two-stage training approach (KCG initialization and global fine-tuning) to potential bias in the initial KCG. While a severely biased initial KCG could theoretically limit the solution space, our design incorporates global fine-tuning that mitigate this risk and promote convergence toward a robust optimum.  the global fine-tuning stage does not freeze the expert parameters (KCG). All the adapter parameters are updated via backpropagation during global SGD. This end-to-end training allows the model to: Adjust the parameters of the KCG to better align with the integrated task objectives; Refine how information flows between experts based on the global loss signal. As validated in [1], **the two-stage training manner outperforms other training methods across various tasks**. Meanwhile, we further conduct two-stage training on MixLoRA and HydraLoRA, both exhibits improvement:
>
>   |  | Training Setting | ID | OOD |
>   | :--- | :--- | :--- | :--- |
>   | MixLoRA | two stage | 76.4 | 63.0 |
>   |  | one stage | 76.0 | 61.0 |
>   | HydraLoRA | two stage | 77.2 | 63.4 |
>   |  | one stage | 77.3 | 62.9 |
>   | HiMoLE | two stage | 77.9 | 65.3 |
>   |  | one stage | 61.7 | 53.5 |
>
> [1]Branch-Train-MiX: Mixing Expert LLMs into a Mixture-of-Experts LLM
>
> [2]HydraLoRA: An Asymmetric LoRA Architecture for Efficient Fine-Tuning
>
> [3]Exploring Generalization in Deep Learning
>
> [4]Evaluating Cross-Domain Named Entity Recognition

---

> ### Comment · Reviewer_Q7d6 · 2025-08-06
>
> Thanks for the detailed rebuttal. I've gone through your answers, the reviewer discussions, and the new results. I think you've mostly addressed my concerns. The additional info is definitely helpful, and I appreciate your effort.
>
> Minor points: (1) the extreme OOD results (W5) showed some improvement, but more explanation behind those findings could help. (2) ablation study could definitely be expanded further to provide more direct results to substantiate design choices.

---

> > ### Author Response · Authors · 2025-08-07
> >
> > Thank you for your constructive feedback. We address the minor points as follows:
> > - Minor points 1: Thank you for highlighting the need for deeper interpretation of the extreme OOD (W5) results. We hypothesize that the observed improvement stems from the domain-specific knowledge intensity: The political domain is inherently less knowledge-intensive compared to biomedicine, where specialized terminology and structured ontologies dominate. This lower knowledge density reduces the reliance on domain-specific priors, allowing the model’s generalizable extraction capabilities—enhanced via supervised fine-tuning —to more effectively generalize to unseen political content. The following table exhibits cases where the model after SFT can correct the original errors in entity labeling.
> >
> >   While these mechanisms partially explain the W5 results, we acknowledge that the causal interplay between task-specific knowledge intensity and cross-domain generalization remains an open question. A systematic investigation of this relationship is reserved for future work.
> >
> >   |  base | sft  | ground truth |
> >   | :--- | :--- | :--- |
> >   |[('country', 'serbia'), ('election', 'serbian presidential election'), ('political party', 'democratic party of serbia'), ('political party', 'list for sandžak'), ('political party', 'new serbia'), ('political party', 'united serbia'), ('politician', 'ilić')] |  [('election', 'serbian presidential election'), ('political party', 'democratic party of serbia'), ('political party', 'list for sandć'), ('political party', 'new serbia'), ('political party', 'united serbia'), ('politician', 'ilić')] | [('election', '2008 serbian presidential election'), ('organization', 'list for sandžak'), ('political party', 'democratic party of serbia'), ('political party', 'new serbia'), ('political party', 'united serbia'), ('politician', 'ilić')] |
> >   |[('else', 'bill'), ('political party', 'labour party'), ('political party', 'liberal democrats'), ('political party', 'plaid cymru'), ('political party', 'scottish national party'), ('political party', 'social democratic and labour party')] | [('political party', 'labour party'), ('political party', 'liberal democrats'), ('political party', 'plaid cymru'), ('political party', 'scottish national party'), ('political party', 'social democratic and labour party')]| [('political party', 'labour party'), ('political party', 'liberal democrats'), ('political party', 'plaid cymru'), ('political party', 'scottish national party'), ('political party', 'social democratic and labour party'), ('politician', 'bill')] |
> >   |[('political party', 'european parliament'), ('political party', 'green party of england and wales'), ('political party', 'plaid cymru'), ('political party', 'uk independence party')]| [('organization', 'european parliament'), ('political party', 'green party of england and wales'), ('political party', 'plaid cymru'), ('political party', 'uk independence party')]|[('organization', 'european parliament'), ('political party', 'green party of england and wales'), ('political party', 'plaid cymru'), ('political party', 'uk independence party')]|
> > - Minor points 2: HiMoLE introduces a novel framework that integrates sentence-level routing with token-level routing in a hierarchical manner. To rigorously evaluate the contribution of HiMoLE’s hierarchical routing design, we compared three routing strategies: Sentence-level routing only, Token-level routing only, Hierarchical routing(HiMoLE).  Results below proves our hierachical method achieves the best performance on both in-distribution and out-of-distribution data. This directly validates that combining sentence- and token-level routing provides complementary benefits between the ID and OOD performance.
> >   |  | ID1 | ID2 | ID3 | ID4 | OOD |
> >   | :--- | :--- | :--- | :--- | :--- |:--- |
> >   | Sentence Routing | 84.4	|58.1	|78.5	|74.1	|64.6|
> >   | Token Routing| 87.4	| 60.6| 	73.4	| 77.3| 	61.0|
> >   | Hierachical Routing| **87.6** | **64.0** | **75.3** | **79.6** |**65.3**|

---

> > > ### Comment · Reviewer_Q7d6 · 2025-08-09
> > >
> > > Thank you for the additional clarification.

---

### Official Review · Reviewer_CeY4 · 2025-07-02

**Clarity:** 3
**Significance:** 2
**Originality:** 3
**Rating:** 4
**Confidence:** 4

**Summary:**

This paper finds that vanilla LoRA and token-level MoE LoRA suffer from limited generalization under out-of-distribution (OOD) conditions. And this paper finds that sentence-level routing improves OOD robustness. Thus, it proposes HiMoLE, a hierarchical expert architecture which combines token-level MoE LoRA and sentence-level MoE LoRA, to enhance in-distribution performance while also ensuring robustness to out-of-distribution data. Moreover, it proposes a two-stage training strategy to enhance performance.

**Questions:**

For HiMoLE, could the authors clarify whether the maximum of 10,000 training steps refers to the combined total of both stage 1 and stage 2, or if it is exclusive to stage 2?

**Ethical Concerns:**

["NO or VERY MINOR ethics concerns only"]

**Final Justification:**

The author has mostly resolved my concerns, especially _weakness 1_ that I was most worried about.

**Limitations:**

HiMoLE appears to be limited to knowledge-intensive tasks, and there is no experimental evidence showing that it can be extended to reasoning-intensive tasks such as math and code.

**Paper Formatting Concerns:**

The checklist should be placed after the appendix.

**Quality:**

3

**Strengths And Weaknesses:**

Strength:
1. The motivation of this paper is relatively clear: token-level routing achieves better in-distribution (ID) performance but struggles to generalize to out-of-distribution (OOD) data. In contrast, sentence-level routing improves OOD robustness at the expense of ID accuracy. To address this trade-off, the paper proposes a hierarchical expert architecture and routing strategy that combines both token-level and sentence-level routing within a PEFT framework.

2. This paper provides a theoretical explanation for why HiMoLE works, from the perspective of gradient conflict reduction enhancing model generalization.

Weakness:

1. Non-zero LoRA initialization can accelerate the training speed of LoRA [1]. The initialization of stage 1 may be the main factor contributing to HiMoLE, rather than the hierarchical expert architectures and routing strategies. The authors should include baseline comparisons that combine stage 1 initialization strategies with MixLoRA and HydraLoRA.

2. Some ablation experiments should be added, such as comparing HiMoLE with the baseline under different LoRA ranks.

3. It seems a bit unusual to me that the experiments use OneKE-13B for the NER task, while Llama2-7B is used for SA and EQA tasks. I wonder why a single model isn't used across all experiments.

4. HiMoLE appears to be limited to knowledge-intensive tasks, and there is no experimental evidence showing that it can be extended to reasoning-intensive tasks such as math and code.

[1] PiSSA: Principal Singular Values and Singular Vectors Adaptation of Large Language Models. In NeurIPS 2024.

---

> ### Author Rebuttal · Authors · 2025-07-31
>
> Thank you for your thoughtful feedback and constructive suggestions. Below, we address the concerns raised:
> - W1: Impact of Stage 1 Initialization
>
>   We appreciate the observation regarding the potential influence of non-zero LoRA initialization. To disentangle the effects of initialization from our hierarchical architecture, we add baseline comparisons where MixLoRA and HydraLoRA are trained with the same Stage 1 initialization as HiMoLE on NER task. Results is as follows, the metric used in the table is F1:
>
>   |  | Training Setting | ID | OOD |
>   | :--- | :--- | :--- | :--- |
>   | MixLoRA | two stage | 76.4| 63.0|
>   |  | one stage | 76.0 | 61.0 |
>   | HydraLoRA | two stage | 77.2 | 63.4 |
>   |  | one stage | 77.3 | 62.9 |
>   | HiMoLE | two stage | **77.9** | **65.3** |
>   |  | one stage | 61.7 | 53.5 |
>
>   Although other methods demonstrate improvements, the two-stage training approach combined with the hierarchical mixture of LoRA experts still delivers **the best performance across both in-distribution and out-of-distribution settings**, with essential improvement under OOD setting.
> - W2: Ablation Studies on LoRA Ranks
>
>   We agree that LoRA rank analysis is critical. We conduct experiments comparing HiMoLE with baseline methods using another LoRA rank(4) on the Sentiment Analysis (SA) task. As shown in the table below, HiMoLE achieves the best performance across different LoRA ranks, with a particularly significant margin(30.1% -> 59.4%) under the OOD setting, further validating the effectiveness of our method.
>
>   | Accuracy(%)| HiMoLE | MixLoRA | HydraLoRA |
>   | :--- | :--- | :--- | :--- |
>   | ID | 72.7 | 69.5 | 69.2 |
>   | OOD | **59.4** | 23.0 | 30.1 |
> - W3: Model Selection Across Tasks
>
>   The choice of different base models (OneKE-13B for NER, Llama2-7B for SA/EQA) was driven by **task requirements**:
>   - OneKE-13B: Specializes in entity-centric knowledge, making it ideal for NER and easy to adapt to medical NER domain.
>   - Llama2-7B: Strong in general language understanding, suitable for SA and EQA.
>   We note that HiMoLE’s architecture is model-agnostic, and its benefits hold across base models.
> - W4 & L1: Generalization to Reasoning-Intensive Tasks
>
>   Thank you for this valuable observation regarding the current scope of HiMoLE. We acknowledge that the experiments presented in the paper primarily focus on knowledge-intensive tasks involving domain shifts and factual robustness, as this is based on the former study [1] and aligns with our core design motivation: addressing the critical challenge of OOD generalization for knowledge-dependent capabilities. This focus is supported by prior research [2,3] indicating that **knowledge-intensive tasks often exhibit more severe OOD generalization degradation compared to reasoning-intensive tasks**, where LLMs tend to demonstrate stronger inherent generalization abilities. Consequently, methods specifically targeting knowledge robustness, like HiMoLE, are particularly relevant for this class of problems.
>
>   Regarding reasoning-intensive tasks (e.g., math, code),  the state-of-the-art performance on complex reasoning tasks often leverages sophisticated techniques like **verifier-guided reinforcement learning** [4], which operates under a different paradigm than the SFT-based PEFT methods central to our current HiMoLE investigation. Building upon this foundation and your insightful comment, our future work will actively explore integrating HiMoLE's principles with  reinforcement learning to enhance LLM's reasoning skills .
> - Q1: Training Step:
>   It refers to the combined total of both stage 1 and stage 2 for HiMoLE.
>
> References:
>
> [1] Revisiting Out-of-distribution Robustness in NLP: Benchmarks, Analysis, and LLMs Evaluations
>
> [2] Generalization vs memorization: Tracing language models’ capabilities back to pretraining data
>
> [3] Quantifying generalization complexity for large language models.
>
> [4] DeepSeek-R1: Incentivizing reasoning capability in LLMs via reinforcement learning

---

> > ### Comment · Reviewer_CeY4 · 2025-08-01
> >
> > Thanks for the author's reply. I increase my rating to 4.

---

> > > ### Author Response · Authors · 2025-08-07
> > >
> > > Thank you for your constructive feedback.

---

### Official Review · Reviewer_aQXK · 2025-07-05

**Clarity:** 3
**Significance:** 2
**Originality:** 1
**Rating:** 4
**Confidence:** 3

**Summary:**

Authors propose a method for enhancing OOD prediction using MoE and PEFT. Their core idea is to use sentence level and token level routing for improving expert selection. The queries are routed to N group of M LoRA experts, done through two FFD networks.

The method is evaluated in three datasets, and compared to a base un-finetuned model, the simple LoRA, and two other variations of LoRA. A set of experiments are also reported on the selection of the experts and also on the effectiveness of the training algorithm.

**Questions:**

I have listed a list of references. Please let me know why they were not discussed or included as baseline models

**Ethical Concerns:**

["NO or VERY MINOR ethics concerns only"]

**Final Justification:**

I increased my review score from 3 to 4. Please add the discussion regarding those models to the paper.

**Limitations:**

Yes

**Quality:**

2

**Strengths And Weaknesses:**

**Strengths:**

-  The paper is easy to read
-  The experiments are detailed
-  The method is well explained, illustrated, and theoretically discussed

**Weaknesses:**

- Relevant work to OOD and PEFT [1,2,3] is not discussed, other related work to MoE is missing [4]. These are neither discussed nor used as baselines. Please correct me if I am wrong.
- The results are not good. In NER, the base OOD model is working better. In SA, on one metric the base OOD model is better. In EQA, the model is working almost identical to the other baselines.
- The core idea of the model (the sentence level and token level routing) is not new [5]

[1] Probing Out-of-Distribution Robustness of Language Models with Parameter-Efficient Transfer Learning, 2023

[2] How Does Fine-Tuning Impact Out-of-Distribution Detection for Vision-Language Models, 2023

[3] SAFT: Towards Out-of-Distribution Generalization in Fine-Tuning, 2024

[4] Mixture of LoRA Experts, 2024

[5] Beyond Distillation: Task-level Mixture-of-Experts for Efficient Inference, 2021

---

> ### Author Rebuttal · Authors · 2025-07-31
>
> Thank you for your thoughtful review and insightful comments. We hereby address your concerns below:
> - W1 & Q1: Relevant work to OOD and PEFT
>
>   Due to space constraints of the main paper, we prioritized foundational and highly influential works in OOD generalization and  LoRA while adopting state-of-the-art Mixture of Parameter-Efficient Experts (MoPE) methods as baselines. Below, we clarify the reason:
>
>   1. References [1-2]: These works focus on OOD detection (e.g., model confidence calibration and thresholding for identifying distribution shifts), whereas our work addresses OOD generalization (improving robustness to unseen domains). Specifically:
>     - [1] investigates the correlation between OOD detection performance and pretrained language model (PLM) scale.
>     - [2] studies CLIP-based fine-tuning’s impact on OOD detection in few-shot settings.
>   By contrast, we diagnose OOD generalization limitations in LoRA-based methods and propose a novel hierarchical MoPE framework to alleviate these issues.
>   2. Reference [3]: This CLIP-based method implements sparsity via a masking strategy, which differs fundamentally from our modular, hierarchical design tailored for LoRA domains. Our approach optimizes parameter efficiency and generalization through dynamic expert activation, rather than masking predefined components.
>   3. Reference [4]: only  introduce the MoPE framework to the FFN layer of every Transformer block for all MoPE methods, while [4] integrates the MoPE framework to the entire Transformer layer. HiMoLE and the baselines included in the paper focus on the hierarchical routing strategy and hierarchical experts designwithin each layer, whereas [4] emphasizes architectural design across layers. Extending HiMoLE to full Transformer layers is planned for future work.
> - W2: The results are not good.
>
>   As discussed in Section 3.1 ("Identification of OOD Generalization Problem in PEFT") and illustrated in Figure 1(a-c), **standard LoRA exhibits significant performance degradation on OOD datasets** when applied to tasks requiring adaptation across diverse knowledge domains. This degradation manifests in **two key ways**: (1) the model performs worse than
>   its pre-fine-tuned counterpart, and (2) the performance gain on OOD data is substantially smaller than that on ID data. To address these limitations, we propose HiMoLE to **mitigate the OOD generalization issue**.
>   While HiMoLE still incurs a performance drop on OOD datasets compared to the base model, it outperforms other LoRA-based methods, achieving up to **5.0%** absolute improvement over the best baseline under OOD settings. When the LoRA rank is set to 4, HiMoLE achieves a substantial improvement on the SA task under the OOD setting, increasing **from 30.1% to 59.4%**.This demonstrates its enhanced robustness to distributional shifts.
>   For the Extractive Question Answering (EQA) task, which exhibits similar challenges, we further conduct statistical significance tests to validate the consistency of HiMoLE’s performance gains. Specifically, we use the average of EM and ROUGE-2 as our evaluation metric and conduct a t-test. The results show that HiMoLE’s gains are statistically significant, with **p-values of 1.8×10⁻⁵ on in-distribution data and 2.3×10⁻⁶ on out-of-distribution data**, reinforcing its reliability in diverse OOD scenarios.
> - W3: The core idea of the model is not new
>
>   While sentence-level and token-level routing are established techniques, our key innovation is a hierarchical framework that synergistically unifies them to overcome the shortcomings of LoRA and earlier MoE methods under distribution shifts. In contrast, **existing MoE approaches either rely on mutually exclusive routing [5] without integration or restrict themselves to token-level routing only [6,7,8]**. As reviewer ADXP mentioned, **vulnerability of LLMs to distribution shift in the data** is an important problem and **yet less explored in the literature**. Specifically:
>   1. We first reveal a key trade-off between routing granularities in Section 3.1: Token-level routing delivers top in-distribution accuracy by using fine-grained context, but it overfits and performs poorly out-of-distribution. Sentence-level routing is more robust out-of-distribution through its focus on overall meaning, yet it loses in-distribution accuracy by overlooking local details.
>   2. To improve performance on both in-distribution (ID) and out-of-distribution (OOD) data, we propose a Hierarchical Routing Strategy that goes beyond the mutually exclusive routing in [5] and the purely token-level routing of existing LoRA MoE methods [6,7,8], including a two-stage cascaded mechanism:
>   - Stage 1 (Sentence-Level Routing): Dynamically assigns input sentences to specialized LoRA experts (In our work we refer it as Knowledge Cooperation Group(KCG)) based on global semantic patterns, ensuring robust OOD generalization.
>   - Stage 2 (Token-Level Routing): Within each KCG, fine-grained token-level gating further adapts local feature representations, preserving ID task precision.
>
> References:
>
> [1] Probing Out-of-Distribution Robustness of Language Models with Parameter-Efficient Transfer Learning, 2023
>
> [2] How Does Fine-Tuning Impact Out-of-Distribution Detection for Vision-Language Models, 2023
>
> [3] SAFT: Towards Out-of-Distribution Generalization in Fine-Tuning, 2024
>
> [4] Mixture of LoRA Experts, 2024
>
> [5] Beyond Distillation: Task-level Mixture-of-Experts for Efficient Inference, 2021
>
> [6] HydraLoRA: An Asymmetric LoRA Architecture for
> Efficient Fine-Tuning
>
> [7] MIXLORA: Enhancing Large Language Models
> Fine-Tuning with LoRA-based Mixture of Experts
>
> [8] LoRAMoE: Alleviate World Knowledge Forgetting in Large Language Models via MoE-Style Plugin

---

> ### Comment · Reviewer_aQXK · 2025-08-06
>
> I increased my review score. Please add the discussion regarding those models to the paper.

---

> > ### Author Response · Authors · 2025-08-07
> >
> > Thank you for your constructive feedback. We appreciate your suggestion and will incorporate a detailed discussion of these models in the revised manuscript. This addition will provide a more comprehensive comparison and better contextualize our findings within the broader research landscape.

---

### Official Review · Reviewer_ADXP · 2025-07-21

**Clarity:** 3
**Significance:** 3
**Originality:** 3
**Rating:** 5
**Confidence:** 3

**Summary:**

This paper is concerned with OOD perfomance of the LoRA  finetuning frameworks. The OOD performance of LoRA is  first shown  empirically to be limited and then  by means of a  mixure-of-experts scheme the  OOD performance is improved.

**Questions:**

- The nuance parameter ‘k’ in eq(4), how do you choose it? have you performed any  optimization on it to find the optimal value? same for $\alpha$ and $\beta$  in the loss function.
- You have tested your  scheme only on LoRA family, while there are other  PEFT methods which have equivalent performance with fewer trainable params such as BitFit, RoCoFT, (IA)3. Is  your method   extendible  directly to other PEFT methods?

**Ethical Concerns:**

["NO or VERY MINOR ethics concerns only"]

**Final Justification:**

The Authors sufficiently addressed my comments and made it cleat why LoRA is chosen and MoE framework for other PEFT  methods while  doable is not trivial.

**Limitations:**

The HiMoLE  scheme  introduced in this paper is only applicable on LoRA family PEFT.

**Paper Formatting Concerns:**

Please  proofread the paper for grammatical  error and typos such as ‘-’ in table3 and capitalization in tables 2-3, and ‘Let g denotes the gradient,…’.  please coorect the formulations such as \theta_i in (15).

**Quality:**

2

**Strengths And Weaknesses:**

Vulnerability of  LLMs  to  distribution shift in the  data is  an important problem and yet less explored in the  literature, therefore this paper is of high interest to the community. The strength is showing theoretically that  hierarchical routing reduces the prevalence of conflicting gradients by inducing structural sparsity in expert selection and this is directly connected to generalization error.

- in (2)-(3) please mention the  sizes of matrices W,h,G, for better readablity.
- please proofread the paper for grammatical errors: "layer replaced the traditional FFN layer can be.."
- the connection between  (10) and (11) is  related to lemma 4 in the appecdix. Not including it in the main text is like a missing part. In particular, how SimGrad and V(.) are connected should be clearly discussed in the main text.
- In algorithm1,  are the parameters  Alpha,beta ,k   fixed and  fed in as the inpute? please  correct this.
-In the proof of lemma 4, there seems to be  some  approximation which  should be denoted by ~ and not =.

---

> ### Author Rebuttal · Authors · 2025-07-31
>
> Thank you for your thoughtful review and insightful comments. We hereby address your concerns below:
> - W1: better readablity of the the sizes of matrices.
>
>   Thanks for the suggestion, we will add in the revised manuscript. The size is as follows: W ∈ ℝn×d; h ∈ ℝd; G ∈ ℝn.
> - W2: grammatical errors
>
>   The sentence "layer replaced the traditional FFN layer can be..." has been revised to: "The layer replacing the traditional FFN layer can be...".
> - W3: missing connection in the main text
>
>   Sorry for the inconvenience, we will add in the revised manuscript.
> - W4: grammatical errors
>
>   Yes, the parameters alpha,beta ,k are fixed. We will correct the representations in the revised manuscript.
>
> - Q1: The configuration of alpha, beta ,k
>   We selected the optimal k based on the Silhouette Coefficient which evaluates the quality of clustering results. The hyperparameters α and β were tuned on a validation set: we tried several values (α, β ∈ {0.1, 0.01, 0.001}) and chose the combination that achieved the best performance on the validation set.
> - Q2 & L1: application to other PEFT methods
>
>   Thank you for raising this important point. While our current implementation of HiMoLE (Hierarchical Mixture of LoRA Experts) is tailored to the LoRA family, this design choice stems from the **inherent compatibility between Mixture-of-Experts (MoE) architectures and LoRA’s low-rank adaptation mechanism**. As shown in recent work ( [1]), **most existing MoE-based parameter-efficient fine-tuning (MoPE) frameworks, including ours, leverage LoRA due to its modularity and ease of integration with sparse expert routing**.
>   The hierarchical routing mechanism in HiMoLE relies on dynamically combining task-specific LoRA adapters. Direct extension to methods like BitFit or (IA)³ would require re-engineering their parameter update rules to align with MoE’s sparse activation paradigm, which is non-trivial. For instance, BitFit updates only bias terms, while (IA)³ introduces learned scaling vectors—neither naturally supports the "expert" abstraction central to MoE.
>   Despite this limitation, our work **diagnoses a critical, underexplored issue: the out-of-distribution (OOD) generalization vulnerability of existing LoRA and MoE-LoRA methods under distribution shift**. We propose a **novel hierarchical MoPE framework to mitigate gradient conflicts via structural sparsity, advancing the theoretical understanding of MoE’s role in LoRA robustness** .We appreciate the reviewer’s insight and agree that broader compatibility with PEFT methods is an exciting direction. Our work prioritizes rigor in addressing LoRA’s OOD limitations while laying a foundation for future MoPE frameworks.
>
> References:
>
> [1]A Survey on Mixture of Experts in Large Language Models

---

> ### Comment · Reviewer_ADXP · 2025-08-02
>
> Thank you for addressing my comments.

---

> > ### Author Response · Authors · 2025-08-07
> >
> > Thank you for your constructive feedback.

---

### Decision · Program_Chairs · 2025-09-17

**Decision:**

Accept (poster)

**Comment:**

This paper introduces a framework (HiMoLE) to improve the out-of-distribution (OOD) robustness of LoRA. HiMoLE proposes a hierarchical mixture of LoRA experts and includes sentence-level and token-level routing. Authors also provides a specialized two-stage algorithm to train this model. As highlighted by the reviewers, the problem studied in the work is important and clearly motivated, and the empirical evidence is promising.

Reviewer CeY4 worried that the performance gains were due to the two-stage initialization rather than the architecture; the authors addressed this by providing new ablation study. The same reviewer also highlighted that authors do not study reasoning tasks and the authors concurred that HiMoLE is motivated with knowledge tasks and it may not work for reasoning tasks. Reviewer aQXK questioned the novelty and results, but was convinced after the authors clarification. Reviewer Q7d6 called for deeper analysis of overhead, hyperparameters, and more extreme OOD settings, all of which the authors provided with new experiments. Finally, Reviewer ADXP's concern about applicability beyond LoRA was addressed by explaining the unique architectural compatibility, which the reviewer accepted. Authors are highly encouraged to include these in the next revision.